# Synthesizing Programmatic Reinforcement Learning Policies with Large Language Model Guided Search

**Max Liu**[1*]   **Chan-Hung Yu**[1*]   **Wei-Hsu Lee**[1]   **Cheng-Wei Hung**[1]
**Yen-Chun Chen**[2]   **Shao-Hua Sun**[1]
[1]National Taiwan University   [2]Microsoft

## ABSTRACT

Programmatic reinforcement learning (PRL) has been explored for representing policies through programs as a means to achieve interpretability and generalization. Despite promising outcomes, current state-of-the-art PRL methods are hindered by sample inefficiency, necessitating tens of millions of program-environment interactions. To tackle this challenge, we introduce a novel LLM-guided search framework (LLM-GS). Our key insight is to leverage the programming expertise and common sense reasoning of LLMs to enhance the efficiency of assumption-free, random-guessing search methods. We address the challenge of LLMs' inability to generate precise and grammatically correct programs in domain-specific languages (DSLs) by proposing a Pythonic-DSL strategy – an LLM is instructed to initially generate Python codes and then convert them into DSL programs. To further optimize the LLM-generated programs, we develop a search algorithm named Scheduled Hill Climbing, designed to efficiently explore the programmatic search space to improve the programs consistently. Experimental results in the Karel domain demonstrate our LLM-GS framework's superior effectiveness and efficiency. Extensive ablation studies further verify the critical role of our Pythonic-DSL strategy and Scheduled Hill Climbing algorithm. Moreover, we conduct experiments with two novel tasks, showing that LLM-GS enables users without programming skills and knowledge of the domain or DSL to describe the tasks in natural language to obtain performant programs.

## 1 INTRODUCTION

Deep reinforcement learning (DRL) has achieved great success from beating the world champion in Go (Silver et al., 2016) to powering the frontier natural language assistants (Ouyang et al., 2022; Bai et al., 2022), and demonstrated great potential in robotics (Jain et al., 2024), autonomous vehicles (Wang et al., 2023b), and recommendation systems (Chen et al., 2023). However, approximating a policy using a deep neural network makes the decision-making process a black box and, therefore, less interpretable and trustable to human users (Heuillet et al., 2021). Subsequently, carefully designed tools and costly human intervention are often required to deploy trustworthy DRL systems (Doshi-Velez and Kim, 2017). Moreover, DRL frequently encounters substantial performance declines when applied to previously unseen scenarios (Kirk et al., 2023; Cobbe et al., 2018), revealing another aspect of its challenges in generalization.

Recent programmatic reinforcement learning (PRL) methods have explored representing an RL policy using a program (Andre and Russell, 2001; Verma et al., 2018; Bastani et al., 2018; Silver et al., 2020). Instead of learning a state-to-action mapping as in DRL, PRL synthesizes programs written in Domain-Specific Languages (DSLs) as human-readable policies that can be parsed and executed, making human inspection possible as an extra safety guard. Moreover, such structured program policies are shown to be able to capture high-level task-solving ideas, allowing for generalizing to a wide range of task variants (Trivedi et al., 2021). Despite the encouraging results, the state-of-the-art PRL algorithms are notoriously inefficient and require tens of millions of program execution in

---

*Equal contribution.   Correspondence to: Shao-Hua Sun <shaohuas@ntu.edu.tw>

environments (Trivedi et al., 2021; Liu et al., 2023; Carvalho et al., 2024). Under the hood, they are search or RL algorithms without any assumption to the targeting problems. On the one hand, this allows generalization to all kinds of problems; on the other hand, the search time grows exponentially with increasing DSL complexity, making it intolerable for any practical use case.

Our key insight is that there is likely only a limited set of problems corresponding to specific program policies that are of human interest – hence, we can utilize reasonable assumptions to prune the search space. Recently, large language models (LLMs) have been demonstrated to possess internet-scale knowledge that can be retrieved by a natural language interface (Wang et al., 2024; Zheng et al., 2024; Taylor et al., 2022). If we view the text on the internet as a "text projection" of human civilization, an LLM as a knowledge base should be able to provide hints to "prune" the program search paths that are out of the "human interest scope". With this intuition, we conjecture LLMs can be utilized to bootstrap the sample efficiency of search-based PRL algorithms, pushing PRL one step closer to practical adoption.

To this end, we aim to develop a PRL framework that utilizes LLMs to produce task-solving program policies while minimizing the number of program executions in environments. Directly instructing LLMs to synthesize DSL programs for solving PRL tasks faces three fundamental challenges: (1) LLMs may lack the domain knowledge of the PRL tasks, *e.g.*, environment dynamics, what an agent can obverse or how it can act, (2) the training data of modern LLMs are mostly natural language texts and general-purpose programming languages, *e.g.*, Python and C++, which can be quite different from the DSLs used in PRL, and (3) there is no apparent mechanism for directly and iteratively optimizing LLMs to produce programs that maximize rewards since the best-performing LLMs are privately owned, *e.g.*, GPT-4 (Achiam et al., 2023).

To combat these challenges, we present an LLM-guided search (LLM-GS) framework leveraging the programming skills and common sense of LLMs and the effectiveness of search algorithms. (1) To familiarize LLMs with PRL tasks, we devise domain and task-aware prompts that convey PRL domain knowledge to LLMs while avoiding leaking task-solving information. (2) To mitigate the gap between general-purpose programming languages and DSLs, we design a Pythonic-DSL strategy that allows LLMs to generate more precise and grammatically correct DSL programs by first producing Python programs. (3) To further optimize the LLM-generated programs, we propose a search algorithm, Scheduled Hill Climbing (Scheduled HC), to efficiently improve programs.

We compare our proposed LLM-GS framework in the Karel domain to various existing PRL methods (Trivedi et al., 2021; Liu et al., 2023; Carvalho et al., 2024). The experimental results demonstrate that LLM-GS is significantly more effective and efficient than the existing methods. Extensive ablation studies show that (1) our proposed Pythonic-DSL strategy leads to a higher ratio of executable (*i.e.* grammatically correct) programs and a higher average return compared to directly generating DSL programs, (2) our proposed search method, Scheduled HC, achieves the best efficiency among existing search algorithms, (3) initializing the search population of Scheduled HC using LLM-generated programs is significantly more efficient than randomly sampled programs. To evaluate whether LLM-GS is useful to users without knowledge of the Karel domain and DSL, we additionally design two novel tasks and only provide LLM-GS with task descriptions while fixing the domain and DSL prompts. The experiment results show that LLM-GS still achieves significantly improved sample efficiency, highlighting the extensibility of our proposed framework.

## 2  RELATED WORK

**Programmatic reinforcement learning (PRL).** PRL represents RL policies using more structured and potentially more interpretable and generalizable representations, such as decision tree (Bastani et al., 2018), state machine (Inala et al., 2020; Koul et al., 2019; Lin et al., 2024), symbolic expression (Verma et al., 2018; 2019; Bhupatiraju et al., 2018; Landajuela et al., 2021), Logic programming language (Jiang and Luo, 2019), and program written in domain-specific language (DSL) (Andre and Russell, 2001; Silver et al., 2020; Sun et al., 2020; Zhu et al., 2019; Qiu and Zhu, 2022; Moraes and Lelis, 2024; Mariño et al., 2021; Trivedi et al., 2021; Liu et al., 2023; Carvalho et al., 2024). Trivedi et al. (2021); Liu et al. (2023) devised PRL tasks in the Karel domain (Pattis, 1981) and proposed learning embedding spaces of programs using variational autoencoders (Kingma and Welling, 2014) and then optimizing program embeddings using cross-entropy method or RL. Carvalho et al. (2024) achieve the state-of-the-art results in the Karel domain (Pattis, 1981) by searching in a programmatic

Program $\rho$ := DEF run m( $s$ m)

Repetition $n$ := 0..19

Perception $h$ := frontIsClear | leftIsClear | rightIsClear |

      markersPresent | noMarkersPresent

Condition $b$ := perception h | not perception h

Action $a$ := move | turnLeft | turnRight |

      putMarker | pickMarker

Statement $s$ := WHILE c( $b$ c) w( $s$ w) | $s_1 s_2$ | $a$ |

      REPEAT R=$n$ r( $s$ r) | IF c( $b$ c) i( $s$ i) |

      IFELSE c( $b$ c) i( $s_1$ i) ELSE e( $s_2$ e)

Figure 1: **The Karel DSL grammar.** It describes the Karel domain-specific language's actions, perceptions, and control flows. The domain-specific language is obtained from Liu et al. (2023).

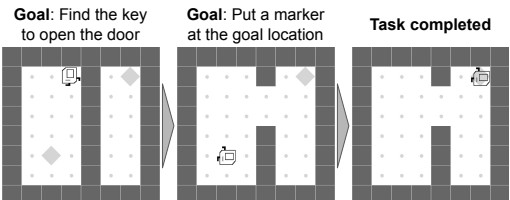

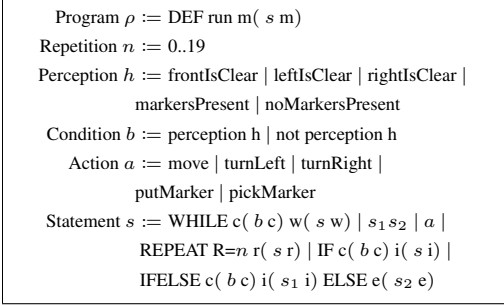

Figure 2: **An example Karel task – DOORKEY.** The agent first needs to find the key (marker) in the left room, which will open the door (wall) to the right room. Navigating to the goal marker in the right room and placing the picked marker on it will grant the full reward for the task. This sparse-reward task has been found to pose significant challenges to previous PRL methods, as it necessitates a greater capability in long-horizon strategy formulation.

space of AST structure. Despite the encouraging results and the generality of these methods, they are notoriously inefficient, requiring tens of millions of program executions to obtain task-solving programs. In this work, to devise an efficient PRL framework, we leverage the knowledge and the reasoning ability of LLMs to generate a set of initial programs to bootstrap search algorithms.

**Large language models for code generation.** With a remarkable ability to understand and generate natural languages and codes, LLMs have been widely adopted for code generation (Brown et al., 2020; Achiam et al., 2023; Nijkamp et al., 2022; Xu et al., 2022; Roziere et al., 2023). These works target general-purpose programming languages for software development, *e.g.* Python and C++, with abundant data on the internet; in contrast, we aim to solve PRL tasks that require writing programs in given domain-specific languages. Prior works exploring using LLMs to synthesize DSL programs via providing LLMs with DSL grammars and few-shot examples, hindsight relabeling, and prioritized experience replay (Wang et al., 2023a; Grand et al., 2024; Butt et al., 2024). While they focus on string/array transformations and abstract reasoning (Chollet, 2019), we leverage LLMs to synthesize DSL program policies to be executed in an RL environment and maximize the return.

**Search-based program synthesis.** Various search algorithms have been developed for program-by-example (PBE) (Gulwani, 2011; Feser et al., 2015; Polozov and Gulwani, 2015; Gulwani et al., 2017; Parisotto et al., 2017; Balog et al., 2017), whose goal is to find programs that satisfy given examples, *e.g.* input/output string pairs. Recent works have explored utilizing search algorithms in DSL to learn libraries (Ellis et al., 2021; Grand et al., 2024). In the regime of programmatic RL, Carvalho et al. (2024) recently applied the Hill Climbing algorithm to the Karel benchmark, achieving state-of-the-art performance. Instead of randomly sampling programs via search algorithms, we present a framework that integrates knowledge from an LLM with a search algorithm to significantly improve the sample efficiency.

An extended discussion on related work can be found in Appendix A.

## 3 PRELIMINARY

**The Karel domain.** We first review the Karel domain, the *de facto* test bed for programmatic reinforcement learning research (Bunel et al., 2018; Chen et al., 2019; Shin et al., 2018; Gupta et al., 2020; Chen et al., 2021). The Karel domain-specific language illustrated in Figure 1 is a robot programming language to control the Karel agent in a 2D grid world. The agent's actions include moving as well as interacting with the environment by picking up and putting down objects (markers). The perceptions check for obstacles and markers, which allows observing the environment. Lastly, control flows, *e.g.*, REPEAT, WHILE, IF, and IFELSE, enable describing complex decision-making logics. More details of the DSL can be found in Appendix B.

We illustrate a Karel task, DOORKEY, in Figure 2. The agent needs to explore the left room using the DSL actions such as move and turnLeft as well as the perceptions such as frontIsClear and markersPresent to find the key (marker). After picking up the key via pickMarker, the wall

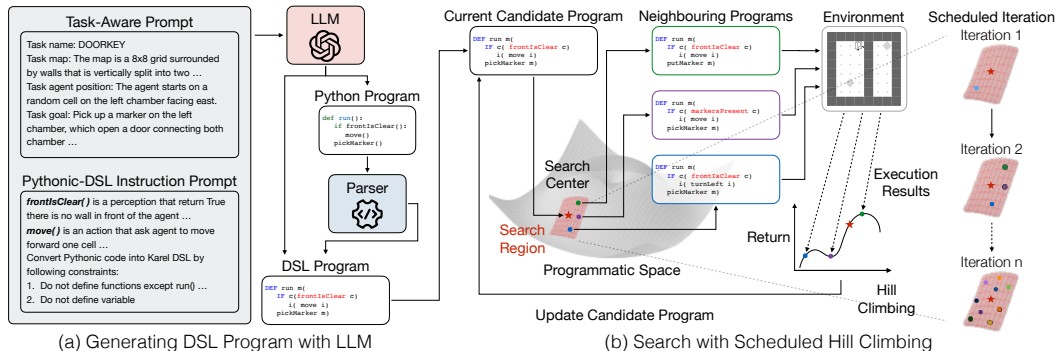

(a) Generating DSL Program with LLM    (b) Search with Scheduled Hill Climbing

Figure 3: **Large language model-guided search (LLM-GS).** (a) With task description and the Pythonic-DSL instruction, LLM generates Python programs that are subsequently converted to DSL programs. (b) These initial programs serve as the initial population of our proposed Scheduled Hill Climbing, which evaluates the episodic return of the neighboring programs to update the current candidate program with increasing neighborhood size over search steps.

to the right room will unblock. The agent then needs to place a marker on the goal marker located in the right room to receive a full task reward. Appendix C presents all the Karel tasks in detail.

The state-of-the-art performance in Karel is achieved by iteratively searching for improved programs according to their episodic return using search algorithms (Carvalho et al., 2024). Next, we introduce the concept of search space, which defines how search algorithms find neighbor programs from a set of current candidates for improved performance, and then dive into the search algorithm details.

**Programmatic space.** In the programmatic space, a program $P$ can be represented as an abstract syntax tree (AST), where each leaf node in the AST represents a program token (Carvalho et al., 2024). To obtain its neighborhood program, a node is sampled from its AST, and then replaced by a subtree randomly generated using the DSL's production rules and sampling strategies.

**Hill climbing (HC).** This search algorithm climbs whenever there is a higher place, *i.e.*, moves to a program having a higher episodic return. Given a program, HC generates its $k$ neighborhood programs from the search space. These programs are evaluated in the environment. If one of them has a higher episodic return than the initial program, it is set as the search center program, and then this process repeats. Otherwise, the algorithm halts and returns the best program evaluated. This search algorithm has a major weakness – it searches surrounding programs in narrow proximity; thus, the distance between initial programs and optimal programs could upper-bound its sample efficiency.

## 4 LARGE LANGUAGE MODEL GUIDED SEARCH

The hill climbing algorithm (HC) is the state-of-the-art programmatic reinforcement learning (PRL) approach in Karel, at the cost of *tens of millions* of program interactions (Carvalho et al., 2024). This prevents its application to real-world decision-making problems, where program-environment interaction at this scale is inapplicable. Hence, we aim to maintain the high episodic return while reducing programs executed in the environments, *i.e.*, *improves the sample efficiency* of PRL. To this end, our key insight is to utilize a large language model (LLM), hypothesizing that its abundant world knowledge, including programming skills and common sense, may bootstrap the inherently assumption-less, random-guessing HC. However, to implement this idea, many challenges rooted in the environment's domain gap (Section 4.1), DSL's language barrier (Section 4.2), and the inability to optimize closed-source LLMs (Section 4.3) must be addressed. Figure 3 presents an overview of our proposed framework, large language model-guided search (LLM-GS).

### 4.1 DOMAIN AND TASK-AWARE PROMPTING

An inexperienced LLM user might directly ask an LLM: "Write a program to solve the Karel task DOORKEY," since this is how typical users interact with LLM-based software development assistants. Obviously, repeating this until the LLM spits out a correct program is unlikely to be more efficient

Table 1: **Activate LLM's domain and task awareness.** To bridge the domain gap between the target task and the LLM's knowledge, we curate a scaffolding prompt to alleviate the knowledge gap and activate the task-solving ability without explicitly dictating the specific programming approach or implementation. Take the task DOORKEY as an example, the user writes down the task name, goal, and some detailed information like map and initial position. Users can easily follow the categories in the prompt to write the task description. The LLM is encouraged to solve the specified task with its programming skills and common sense. More details of the prompts can be found in Appendix D.

| Knowledge | Prompt Text |
| --- | --- |
| Task Name | DOORKEY |
| Map Description | The map is a $8 \times 8$ grid surrounded by walls that is vertically split into two chambers. The left chamber is $6 \times 3$ grid and the right chamber is $6 \times 2$ grid. There is a marker placed randomly on the left chamber as a key, and another marker placed randomly on the right chamber as a goal. |
| Initial Position | The agent starts on a random cell on the left chamber facing east. |
| Task Goal | The goal of the agent is to pick up a marker on the left chamber, which opens a door connecting both chambers. Allow the agent to reach and put a marker on the goal marker. |

than existing search algorithms. There are two major challenges that exist: (1) the LLM lacks the dynamic environmental concepts inherent to the Karel domain, and (2) the LLM does not have background knowledge of the specific PRL task DOORKEY. Therefore, it is crucial to provide a detailed description of the task and environment in natural language so the LLM can make informed assumptions based on its learned common sense.

To address this, we devise a prompting strategy that converts the task description and the task-agnostic environment knowledge into natural language sentences that LLMs can process and reason. The task-agnostic environment knowledge prompt introduces the basic mechanics of interacting with this environment to the model, setting the stage for more specific programming tasks within the Karel framework, detailed in Appendix D.1. Table 1 provides an example of a user prompt for the Karel task DOORKEY. Now that the LLM has acquired basic knowledge of the Karel domain and the task, we can further instruct it to leverage its programming skills, algorithmic knowledge, and long-term planning abilities to generate programs that solve tasks. Note that given this domain-aware prompt, given any novel task within the same domain, even a user without any programming skills can simply describe the task in natural language to obtain a performant program, as shown in Section 5.4.

## 4.2 GENERATING DSL PROGRAMS WITH PYTHONIC-DSL STRATEGY

Given LLMs' widespread success in assisting software development, one might assume an LLM can naively generate DSL programs. Yet, precisely generating DSL programs turns out to be quite challenging for LLMs – their training data typically consists of natural language corpus and general-purpose programming language codes, thus specific DSLs used in PRL tasks are actually quite exotic for them. Note that this language barrier is beyond lexical syntax differences. For example, the DSL may specify rules that limit the usage of temporary variables constrained by the actual robotic hardware, which is rarely a concern for common Python / JavaScript code found on the internet.

Despite the issue, we still hope to leverage the best of LLM's programming skills, algorithmic knowledge, and long-term planning abilities to synthesize DSL programs. To this end, we derive a solution from the following assumptions: (1) The LLM is proficient in a general-purpose programming language, such as Python, (2) The DSL can be represented by a possibly restricted version of the general-purpose programming language, and (3) The LLM understands the restriction in natural language, if any. Our Pythonic-DSL strategy instructs the LLM to generate Python programs instead, given the Karel rules and constraints we wrote in English, and then later convert it into the Karel DSL. As shown in Table 2, DSL details are provided, including the action, perception, primitives and language constraints. Thus, the LLM can generate programs based on its innate general programming skills and its understanding of the DSL description. One caveat is that the LLM still occasionally generates Python programs that cause Python-to-DSL parsing failures. An empirical useful mitigation is instructing the LLM to generate the DSL program converted from its own output Python program

Table 2: **The Pythonic-DSL instruction.** The Karel DSL is specified via a *constrained* version of Python with pre-defined functions, including *Pythonic-DSL perceptions* that allow the agent to observe, and *Pythonic-DSL actions* that enable the agent to interact. This Pythonic-DSL description is fed into the LLM via the system prompt so that it can generate Python code that can later be converted into Karel DSL. We will show that this approach outperforms the direct generation of DSL or Python programs in Table 3. The full prompt is presented in Appendix D.1.

| Knowledge | Prompt Text |
|---|---|
| Pythonic-DSL Perceptions | `frontIsClear()`: Returns `True` if there is no wall in front of the agent. `markersPresent()`: Returns `True` if there exist markers on the current cell. ... *(more perceptions omitted)* |
| Pythonic-DSL Actions | `move()`: Asks the agent to move forward one cell ... *(truncated)* `turnLeft()`: Asks the agent to rotate 90 degrees counter-clockwise. `pickMarker()`: Asks the agent to pick up one marker from the current cell. ... *(more actions omitted)* |
| Language Constraints | - do not define other functions besides `run()` - do not define variables ... *(more rules omitted)* |

as a backup. At this point, we have developed the mechanism to instruct an LLM to generate DSL programs given the PRL task and the DSL grammar in natural language.

### 4.3 OPTIMIZING THE PROGRAM POLICY WITH SCHEDULED HILL CLIMBING

Although the aforementioned techniques can already instruct the LLM to produce DSL programs that solve the target task to a certain degree, the episodic return can still be far from optimal. Since best-in-class LLMs are typically proprietary APIs, directly and iteratively fine-tuning them via gradient-based policy optimization methods is not possible. To further optimize the program generated by the LLM, we explore initializing search populations of search algorithms using LLM-generated programs. Via extensive experiments with the HC, CEM, and CEBS search algorithms on both program space and latent space whenever applicable, we discover a key improvement that is crucial when initializing search from LLM-generated programs – program-environment interaction scheduling for a more efficient allocation of the interaction budget. The intuition is that LLMs can often provide good initialization and therefore it makes sense to keep a small *search budget*, *i.e.* with a smaller population. After a while, if an optimal program is not yet found, we can gradually increase the budget and facilitate broader exploration. We design a scheduler based on this intuition:

$$\log_2 k(n) = (1 - r(n)) \log_2 K_{start} + r(n) \log_2 K_{end}, \tag{1}$$

where $n$ represents the number of evaluated programs, $k(n)$ denotes a function indicating the current number of neighborhood programs based on the number of evaluated programs $n$. This function's logarithm is a linear combination of the logarithms of two hyperparameters, $K_{start}$ and $K_{end}$, which signify the initial and terminal numbers of neighbors to search, respectively. The variable $r(n)$ governs the linear ratio, which itself is a sinusoidal function that gradually increases from 0 to 1 throughout the evaluation of a total of $N$ programs (see Appendix E). With this scheduler, the number of neighbors $k$ is a function of the execution budget used, growing from $K_{start}$ to $K_{end}$ on an exponential basis. We apply this scheduler to Hill Climbing and call this method Scheduled Hill Climbing (Scheduled HC).

Overall, our best recipe consists of the following steps. First, we sample DSL programs for the LLM using the aforementioned techniques. Next, these programs are evaluated in the environment for the episodic return. In some easy tasks, some generated programs might have achieved perfect returns and thus are optimal programs. If no optimal program is found, we sort the programs based on the total reward decreasingly. Next, programs are selected from the sorted list, each serves as an initial program for the HC search, which is the best-performing search algorithm from our extensive experiments. HC searches $k$ neighbors at each step following the scheduler in Eq. (1). This process is repeated until either the optimal program is found or we meet the maximum program-environment interaction budget allowed ($N$).

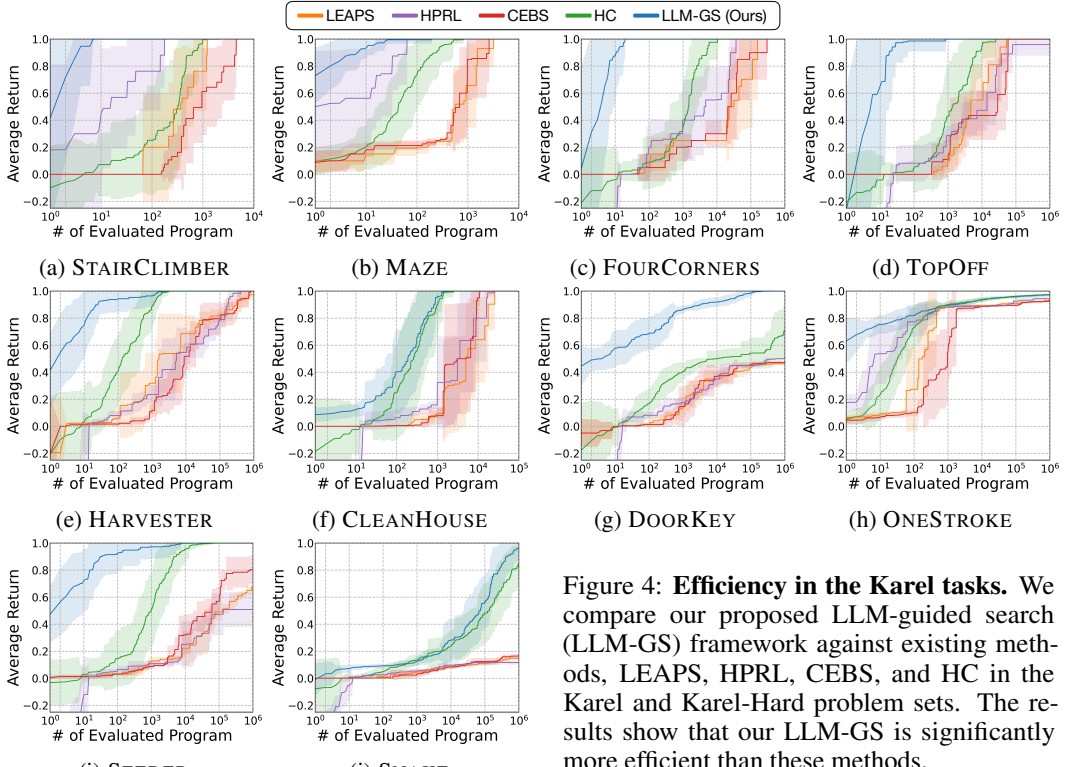

(a) STAIRCLIMBER  (b) MAZE  (c) FOURCORNERS  (d) TOPOFF

(e) HARVESTER  (f) CLEANHOUSE  (g) DOORKEY  (h) ONESTROKE

(i) SEEDER  (j) SNAKE

Figure 4: **Efficiency in the Karel tasks.** We compare our proposed LLM-guided search (LLM-GS) framework against existing methods, LEAPS, HPRL, CEBS, and HC in the Karel and Karel-Hard problem sets. The results show that our LLM-GS is significantly more efficient than these methods.

## 5 EXPERIMENTS

### 5.1 EVALUATION SETUP

**PRL tasks.** We evaluate our proposed framework LLM-guided search (LLM-GS) using the Karel tasks from the two problem sets: Karel (Trivedi et al., 2021) (STAIRCLIMBER, MAZE, FOURCORNERS, TOPOFF, HARVESTER, and CLEANHOUSE) and Karel-Hard (Liu et al., 2023) (DOORKEY, ONESTROKE, SEEDER, and SNAKE). More task details can be found in Appendix C.

**Baselines.** We compare our proposed framework with the following methods: LEAPS (Trivedi et al., 2021), HPRL (Liu et al., 2023), CEBS (Carvalho et al., 2024), and Hill Climbing (Carvalho et al., 2024) (the current state-of-the-art). Note that LEAPS and CEBS search programs in learned latent program space, as introduced in Appendix F.

**Metrics.** For a fair comparison in sample efficiency, we use the number of programs evaluated to represent the program-environment interaction budget, *i.e.* a method is more sample-efficient if it achieves a higher average episodic return on a fixed budget. Specifically, for each task, the number of task variances is $C$. Task variances arise from differences in the environment's initial states. Each program evaluation means executing the program on all $C$ task variances to obtain an average return. We set the number of task variances $C = 32$, the maximum number of program evaluation $N = 10^6$, *i.e.*, the interaction budget. Programs achieving an average return of $1.0$ are considered optimal.

**Setup.** We use GPT-4 (Achiam et al., 2023) (`gpt-4-turbo-2024-04-09` with temperature=1.0, top_p=1.0) as our LLM module to generate the initial search population. The scheduler of our proposed Scheduled HC starts from $K_{start} = 32$ to $K_{end} = 2048$. We evaluate our LLM-GS and HC with 32 random seeds and 5 seeds for LEAPS, HPRL, and CEBS.

**Data Leakage.** When using modern LLMs, a key concern is whether the model has already memorized the content in the test bed, leading to biased or inflated results. We carefully rule out this possibility in Appendix G by examining the LLM release dates and the availability of optimal Karel programs, and by probing the LLM we use.

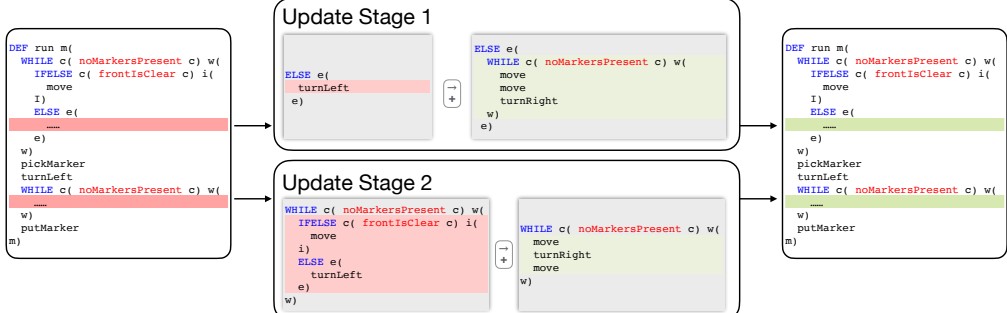

Figure 5: **Example on DOORKEY.** This example shows how our search method improves an LLM-initialized program to an optimal one. The original program (left) has a two-stage structure but lacks navigation ability. The improved program (right) solves this by enhancing its navigating ability on both stages, allowing for solving the task.

## 5.2 LLM-GUIDED SEARCH SIGNIFICANTLY IMPROVES THE SAMPLE EFFICIENCY

Figure 4 presents the experimental result, indicating that our proposed LLM-GS surpasses all the existing methods by a large margin on almost all the tasks.

On the Karel set, our framework completely solves these tasks and exhibits the best sample efficiency. Also, LLM-GS performs extraordinarily well on SEEDER, ONESTROKE, and DOORKEY, which are among the hardest of the Karel-Hard set. We highlight our DOORKEY result, with an efficiency improvement from *not converging at even 1M* (baselines) to *converging within around 500K* (ours). Existing PRL algorithms struggle at DOORKEY due to the fact that it is a two-stage task with a sparse reward function. Therefore, most of the baselines cannot escape the local maxima and converge around $0.5$. Curiously, our method starts at an episodic return of $\approx 0.5$ from the beginning and converges quickly without being trapped by the local maxima. To analyze, we examine one output program from LLM shown in Figure 5. The LLM clearly understands that DOORKEY is a two-stage task and thus generates a corresponding two-stage structured program, which is much easier for later search to find the optimal program. The full dialogue is presented in Appendix D.

We include a plot aggregating the performance across all ten tasks in Appendix H, a wall-clock time (real-time elapsed) evaluation in Appendix I, and a statistical significance test in Appendix J.

## 5.3 ABLATION STUDIES

**Which language to generate, Python or DSL? Do both!** To justify the effectiveness of our proposed Pythonic-DSL strategy, we conduct experiments on generating either Python or DSL programs with the same 8 seeds and present the results in Table 3. The *acceptance rate* calculates the ratio of executable DSL programs, while the *best return* refers to the highest episodic return among the legal programs. Our method achieves the highest acceptance rate on **9 / 10** tasks and the highest best return on **7 / 10** tasks. The complete prompts are presented in Appendix K.

**Our proposed Scheduled HC is the most efficient search method.** We compare various search methods in both programmatic and latent spaces in DOORKEY and CLEANHOUSE using LLM-initialized programs. In the learned latent space, we run CEM, CEBS, HC-250 (HC with a fixed population size of $250$), and our proposed Scheduled HC; in the programmatic space, we run HC-$k \in \{32, 256, 2048\}$ and our proposed Scheduled HC. The results presented in Figure 6 show that HC in the programmatic space achieves the best performance compared to latent space search. However, HC-$k$ with a fixed population size only specializes in specific tasks. To be specific, HC-32 for CLEANHOUSE and HC-2048 for DOORKEY. In contrast, our proposed Scheduled HC achieves comparatively the best performance in both tasks.

**LLM provides a better initial search population.** To verify the efficacy of using LLM-generated programs as search initialization for Scheduled HC, we compare it to using randomly sampled programs as initialization in DOORKEY and CLEANHOUSE. The results shown in Figure 7 suggest

Table 3: **Pythonic-DSL strategy ablation.** We compare the programs generated using our proposed Pythonic-DSL strategy to directly generated Python or DSL programs. Our method achieves the highest acceptance rate and best return, justifying the effectiveness of our Pythonic-DSL strategy.

| Task | Python | | DSL | | Pythonic-DSL (ours) | |
|---|---|---|---|---|---|---|
| | Acceptance Rate | Best Return | Acceptance Rate | Best Return | Acceptance Rate | Best Return |
| STAIRCLIMBER | 88.02±3.57% | **1.00±0.00** | 45.31±11.59% | 0.75±0.65 | **88.54±4.54%** | **1.00±0.00** |
| MAZE | 72.40±7.49% | 0.94±0.08 | 57.81±8.12% | 0.88±0.07 | **92.45±3.89%** | **0.98±0.05** |
| FOURCORNERS | 83.07±5.04% | **1.00±0.00** | 93.49±2.43% | **1.00±0.00** | **94.01±2.43%** | **1.00±0.00** |
| TOPOFF | 90.89±2.07% | 0.97±0.07 | 94.53±4.29% | **1.00±0.00** | **96.35±3.08%** | **1.00±0.00** |
| HARVESTER | 67.45±7.51% | **0.98±0.03** | 79.69±7.64% | **0.98±0.03** | **80.21±5.51%** | 0.95±0.09 |
| CLEANHOUSE | 62.50±3.45% | 0.27±0.28 | 76.30±6.16% | **0.36±0.14** | **84.64±4.29%** | 0.29±0.27 |
| DOORKEY | 90.10±3.86% | **0.63±0.13** | 75.78±15.09% | 0.55±0.05 | **94.53±3.60%** | **0.63±0.14** |
| ONESTROKE | 85.42±4.77% | 0.77±0.06 | 85.68±7.54% | 0.74±0.09 | **95.31±3.08%** | **0.83±0.08** |
| SEEDER | 73.70±4.77% | **0.93±0.14** | 88.28±4.88% | 0.66±0.10 | 74.74±7.10% | 0.90±0.14 |
| SNAKE | 87.50±2.76% | **0.09±0.00** | 30.47±9.02% | **0.09±0.00** | **98.96±1.04%** | **0.09±0.00** |

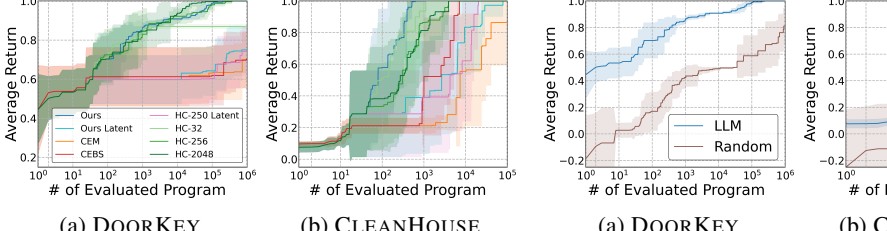

| (a) DOORKEY | (b) CLEANHOUSE | (a) DOORKEY | (b) CLEANHOUSE |

Figure 6: **Comparing search algorithms.** We compare the proposed Scheduled HC with latent space CEM, CEBS, HC-250, and Scheduled HC, and programmatic space HC-$k \in \{32, 256, 2048\}$ using LLM-initialized programs. Our method achieves comparatively good efficiency in DOORKEY and performs best in CLEANHOUSE, justifying the efficacy of our proposed search method.

Figure 7: **Comparing initializations.** We compare initializing the search population using LLM-generated programs or randomly sampled programs, combined with our proposed SHC. The results demonstrate that starting from LLM-generated programs significantly improves efficiency and performance, especially in DOORKEY, highlighting the effectiveness of our proposed framework.

that initializing the search with LLM-generated programs significantly improves the sample efficiency compared to randomly initializing population-like search methods.

**Scheduled HC variants.** We ablate the components in the scheduled HC, *i.e.* logarithmic interpolation, sinusoidal schedule, and logarithmic ratio, and present the results in Appendix E.3.

## 5.4 PERFORMANCE ON NOVEL TASKS SHOWCASES THE EXTENSIBILITY OF LLM-GS

A key advantage of our proposed LLM-GS framework is its ability to adapt to new tasks using only text descriptions without requiring users to have any domain knowledge or familiarity with the DSL. To showcase this feature, we designed two novel Karel tasks, PATHFOLLOW and WALLAVOIDER, as detailed in Appendix C.3. By simply replacing the task descriptions while keeping the Karel domain prompt, Python-to-DSL prompt, and hyperparameters the same, we compare our method with the best-performing baseline, HC. The results presented in Figure 8 show that by only changing task descriptions, LLM-GS still achieves improved sample efficiency and finds better program policies.

## 5.5 PERFORMANCE ON MINIGRID DOMAIN VERIFIES THE ADAPTABILITY OF LLM-GS

To verify if our proposed LLM-GS framework can be adapted to another domain, we adopt three tasks in the Minigrid domain, LAVAGAP, PUTNEAR, and REDBLUEDOOR. Unlike the case in Karel, the perception functions in our Minigrid DSL are parameterized, i.e., require an input parameter such as an object, as detailed in Appendix L. By adapting all aspects of the prompt from the Karel domain to the Minigrid version, including the domain prompt, the task descriptions, and the Python-to-DSL

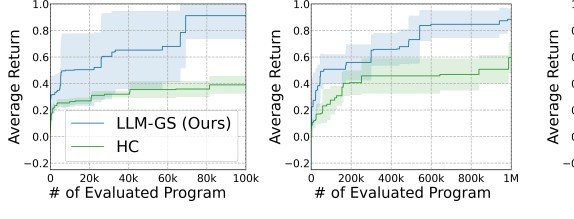

(a) PATHFOLLOW  (b) WALLAVOIDER  (a) LAVAGAP  (b) PUTNEAR

Figure 8: **Performance of PATHFOLLOW & WALLAVOIDER.** With only changing the task descriptions, LLM-GS surpasses the best-performing baseline HC on two novel tasks, highlighting the extensibility of LLM-GS.

Figure 9: **Performance of Minigrid LavaGap and PutNear.** We compare LLM-GS and the best-performing baseline in Karel, *i.e.* HC, in the Minigrid domain. LLM-GS demonstrates better sample efficiency in both of the two tasks.

prompt, we evaluate our method against the strongest baseline, HC. The results presented in Figure 9 show that our proposed LLM-GS is more sample-efficient in the Minigrid domain.

## 5.6 BEYOND INITIALIZATION – A PRELIMINARY ATTEMPT ON LLM REVISION

Given the encouraging results from LLM-initialized programs, one might be curious whether LLM itself without the help of any search algorithms can progressively revise and improve its programs once given feedback from the environment, *e.g.*, reward. For completeness, we additionally conduct studies with the following LLM-based revision:

- **Regeneration.** Inspired by Chen et al. (2024) and Olausson et al. (2024), we instruct the LLM to re-generate non-duplicate programs given its previously generated programs.
- **Regeneration with reward.** In addition to the historical program list, the episodic return of each program is appended to the LLM's inputs.
- **Agent execution trace.** Inspired by prior arts that leverage environment knowledge (Tang et al., 2024; Wang et al., 2024) and code repair (Chen et al., 2024; Olausson et al., 2024), we feed the program with the best episodic return and its execution traces into the LLM.
- **Agent and program execution trace.** Inspired by Hu et al. (2024), we additionally point out the line of the program currently being executed as extra hints to the LLM.

We conduct the LLM revision experiments on DOORKEY for five revision rounds, each round with 32 programs with 5 seeds. The results are presented in Figure 10. We can see that the improvement quickly saturates within the first two rounds, and no significant gain is observed with more revision rounds. This indicates that only using LLM to revise the program may not be sufficient to solve the task. On the other hand, our proposed Scheduled HC can converge to an optimal program – and is "free," *i.e.* no expensive API calls or "API usage has reached your notification threshold" emails. The details can be found in Appendix K.

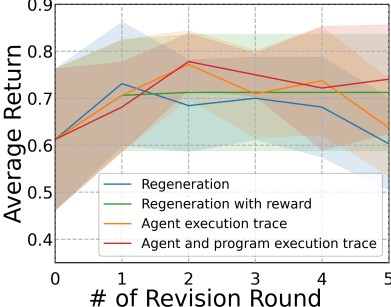

Figure 10: **LLM revision.** The performance gain of LLM revision saturates within a few program revision rounds.

## 6 DISCUSSION

We propose the LLM-guided search (LLM-GS) that leverages the programming skills and common sense of LLM to bootstrap the efficiency of assumption-free, random-guessing search methods. We propose a Pythonic-DSL method to address LLMs' inability to generate precise and grammatically correct programs. To improve the programs generated by LLMs, we design a search algorithm, Scheduled Hill Climbing, which can explore the programmatic search space to consistently improve the programs. The experimental results show the improved sample efficiency of LLM-GS. Extensive ablation studies verify the effectiveness of our proposed Pythonic-DSL method and Scheduled Hill Climbing. The limitations and future directions of LLM-GS are presented in Appendix M.

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

APPENDIX

# Table of Contents

## A    EXTENDED RELATED WORK

**Sample efficiency in programmatic reinforcement learning .** While deep reinforcement learning has achieved tremendous success in various domains (Silver et al., 2016; Ouyang et al., 2022; Bai et al., 2022; Wang et al., 2023b; Chen et al., 2023), it still suffers from sample inefficiency problems. Many methods make their effort to improve sample efficiency including modeling transition model (Kaiser et al., 2019), replay buffer (Mnih et al., 2015), hindsight relabeling (Andrychowicz et al., 2017), ensembling (Wiering and Van Hasselt, 2008), and importance sampling (Sutton, 2018). Likewise, this is also the case for programmatic reinforcement learning (PRL) algorithms. The SOTA algorithm, Hill Climbing (HC), still needs to generate thousands to millions of programs to achieve optimal policies. In this work, we utilize the information on Karel's tasks and LLM programming skills to initiate the search population of our proposed Scheduled HC to improve the sample efficiency. Incorporating other sample efficiency-improving methods into PRL algorithms is a promising research direction, but is orthogonal to our contribution.

**Large language models and search-based program synthesis.** Various search algorithms have been developed for program-by-example (PBE) (Gulwani, 2011; Feser et al., 2015; Polozov and Gulwani, 2015; Gulwani et al., 2017; Parisotto et al., 2017; Balog et al., 2017), whose goal is to find programs that satisfy given examples, *e.g.*, input/output string pairs. Recent works have explored utilizing search algorithms in DSL to learn libraries (Ellis et al., 2021; Grand et al., 2024; Eberhardinger et al., 2023; Wüst et al., 2024) for PBE, imitating policies (Bastani et al., 2018; Kohler et al., 2024), and object representations (Wüst et al., 2024). Neural program synthesis methods (Lin et al., 2018; Sun et al., 2018; Desai et al., 2016; Raza et al., 2015; Wang et al., 2017; Zhong et al., 2023; Le et al., 2022; Parisotto et al., 2017; Balog et al., 2017) learn neural networks from data to synthesize programs intended by users. Grand et al. (2024); Eberhardinger et al. (2023) show that LLMs can bootstrap program synthesis in the wake stage of library learning; it is an open question whether the LLM can bootstrap the sleep stages. We utilize the description of the Karel tasks to bootstrap our scheduled HC. Additional information may be crucial for LLM assist bootstrapping and fine-tuning. This additional information may come from the execution (Chen et al., 2024), prompt design (Wang et al., 2023a; Wei et al., 2022; Wüst et al., 2024; Grand et al., 2024), and the task description in our work.

**More on search methods and structured policy representations.** There are several different search methods developed for programming by example (PBE) (Gulwani, 2011; Feser et al., 2015; Polozov and Gulwani, 2015; Gulwani et al., 2017; Parisotto et al., 2017; Balog et al., 2017) or DSLs with learned libraries (Ellis et al., 2021; Grand et al., 2024; Eberhardinger et al., 2023; Wüst et al., 2024) for PBE, imitating policies (Bastani et al., 2018; Kohler et al., 2024), and object representations (Wüst et al., 2024). Recent studies explore different representations for Reinforcement Learning policies toward interpretable and/or explainable. Together with various structured policy representations such as DSL programs (Andre and Russell, 2001; Verma et al., 2018; Silver et al., 2020), decision trees (Bastani et al., 2018; Kohler et al., 2024), state machines (Lin et al., 2024), and symbolic programs (Jiang and Luo, 2019; Qiu and Zhu, 2022; Delfosse et al., 2024). Our method has the greatest bootstrapping ability for DSL programs but is difficult to extend to all kinds of representations. We designed our prompt by contrasting the policy representation (Karel DSL) with Python. As a result, our prompt in Appendix D.1 limits some of the functionalities of Python. We believe our framework is only suitable for tasks with task-solving procedures that could be described using languages. That said, it could be difficult to apply this framework to low-level control tasks, such as motor torque control (Qiu and Zhu, 2022).

## B    KAREL DSL DETAILS

In our framework, we ask the LLM to write the program in Python and then translate it into DSL, all in one response. We include a sample output from the LLM in Appendix D.3. To translate the responded program to an abstract syntax tree (AST), we use an off-the-shelf Karel parser which can convert a string in DSL into an AST. We implement a translator to convert a Python program into Karel DSL. We list common mistakes LLM made in Table 6 and how we solve these problems.

There are 5 symbols in the Karel DSL: statement, condition, action, boolean, and number. The production rules and Python converting rules are listed in Table 4. The statement counts are the total

statements connected to the root node of the abstract syntax tree (AST), and the statement counts are limited to 6 in search of optimal programs. The program length is the total number of tokens used in string representation and is set to 44. The depth in DSL is the recursive call of control flows, which is limited to 4. These limitations are the same as the original settings in LEAPS (Trivedi et al., 2021). The probabilities of the production rules are listed in Table 5. The mutation process selects one random node in the AST. We use the production rules to sample a sub-tree to replace the original node. The Karel syntax has many symbol-related parentheses, *e.g.*, "i(" and "i)" which are related to the "IF" and "IFELSE", thus LLM sometimes makes mistakes translating Python to DSL.

Table 4: Python to DSL converting rules.

| Python | DSL |
|---|---|
| def run(): s | DEF run m( s m) |
| while b: s | WHILE c( b c) w( s w) |
| if b: s | IF c( b c) i( s i) |
| if b: s else: s | IFELSE c( b c) i( s i) ELSE e( s e) |
| for i in range(n): s | REPEAT R=n r( s r) |
| not h | not c( h c) |
| frontIsClear() | frontIsClear |
| leftIsClear() | leftIsClear |
| rightIsClear() | rightIsClear |
| markersPresent() | markersPresent |
| noMarkersPresent() | noMarkersPresent |
| move() | move |
| turnLeft() | turnLeft |
| turnRight() | turnRight |
| putMarker() | putMarker |
| pickMarker() | pickMarker |

## C  KAREL TASKS

We give the large language model 5 pieces of information related to the tasks: task name, map description, initial Position, task goal, and task return. By converting the original task description into these 5 categories, this design allows the user to easily fill in all categories for new tasks. Here we list all the information pieces for these tasks. Appendix K.1.2 shows how these 5 different components are filled into the placeholders in the user prompt. Appendix D.2 show all filled results with the task DOORKEY. There are three sets of tasks: Karel (Trivedi et al., 2021), Karel-Hard (Liu et al., 2023), and new Karel tasks. Appendix C.1 lists all 6 Karel tasks, Appendix C.2 lists all 4 Karel-Hard tasks, and Appendix C.3 lists the 2 new tasks. Example figures of Karel tasks are in Figure 11, Karel-Hard tasks are in Figure 12 and new Karel tasks are in Figure 13.

Table 5: Probabilities of the production rules.

| Category | Rule | Probability |
|---|---|---|
| **Program P** | Statement | 1.0 |
| **Statement S** | While | 0.15 |
| | Repeat | 0.03 |
| | Concatenate | 0.5 |
| | If | 0.08 |
| | Ifelse | 0.04 |
| | Action | 0.2 |
| **Condition c** | Boolean | 0.9 |
| | not | 0.1 |
| **Action a** | move | 0.5 |
| | turnLeft | 0.15 |
| | turnRight | 0.15 |
| | putMarker | 0.1 |
| | pickMarker | 0.1 |
| **Boolean b** | frontIsClear | 0.5 |
| | leftIsClear | 0.15 |
| | rightIsClear | 0.15 |
| | markersPresent | 0.1 |
| | noMarkersPresent | 0.1 |
| **Number n** | i ($0 \leq i \leq 19$) | 0.05 |

Table 6: These are the common mistakes LLM makes. We implement a simple program to correct these mistakes.

| Type | Before | After |
|---|---|---|
| Brackets removal | DEF run m( move**()** m) | DEF run m( move m) |
| Brackets separation | DEF run **m(**move m) | DEF run **m(** move m) |
| Brackets addition | DEF run m( WHILE frontIsClear w( move w) )m | DEF run m( WHILE **c(** frontIsClear **c)** w( move w) )m |
| Brackets repairment | DEF run m( move ) | DEF run m( move **m)** |
| If to IFELSE | DEF run m( **IF** c( frontIsClear c) i( move i) ELSE e( turnLeft e) )m | DEF run m( **IFELSE** c( frontIsClear c) i( move i) ELSE e( turnLeft e) )m |
| Redundant symbols removal | DEF run m( move() **m** m) | DEF run m( move m) |
| Illegal symbols transformation | DEF run m( WHILE c( True c) w( move w) ) | DEF run m( REPEAT r=19 r( move r) m) |

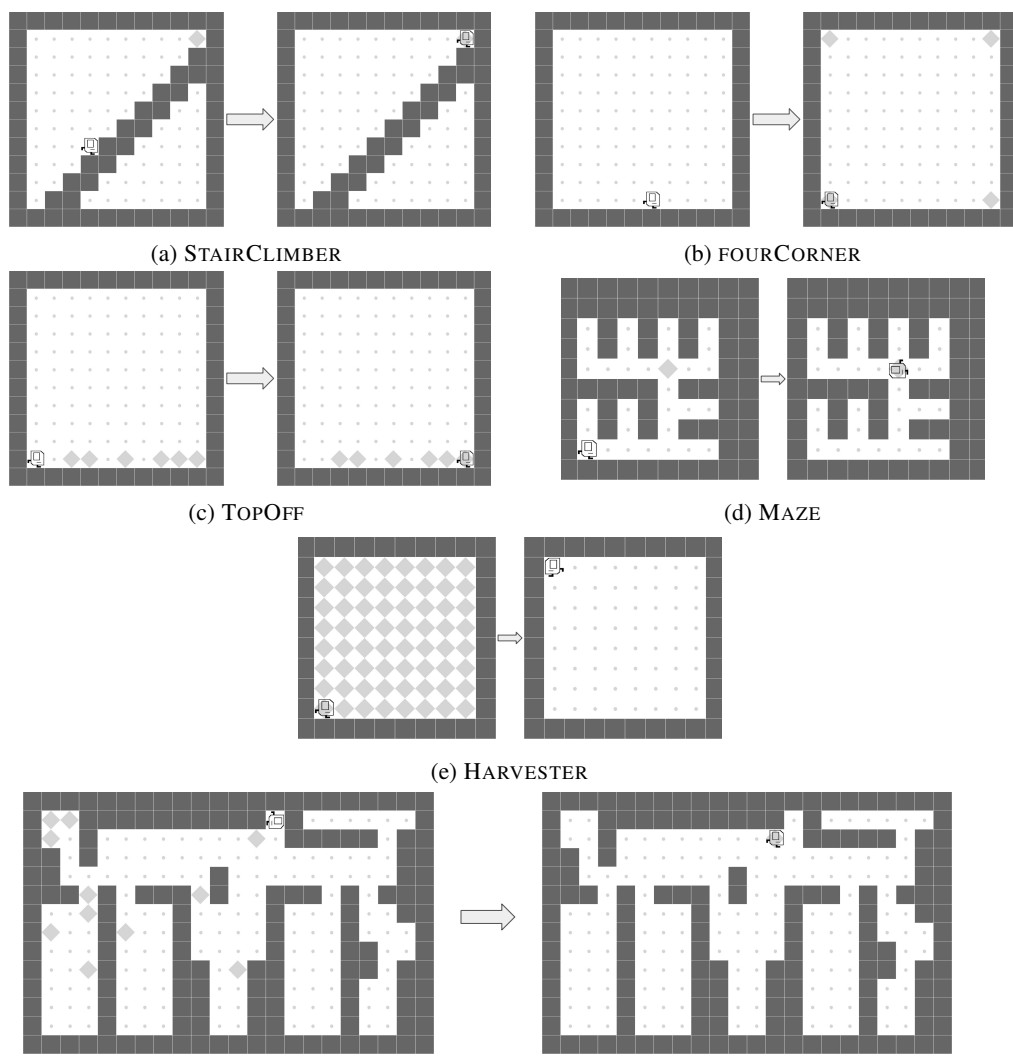

(a) STAIRCLIMBER

(b) FOURCORNER

(c) TOPOFF

(d) MAZE

(e) HARVESTER

(f) CLEANHOUSE

Figure 11: Illustrations of the initial and desired final state of each task in the KAREL Problem set introduced in by Trivedi et al. (2021). Note that these illustrations are from Trivedi et al. (2021) except for STAIRCLIMBER, TOPOFF, and FOURCORNERS. We align our setting with Carvalho et al. (2024) to evaluate these tasks in a map of 12x12. The position of markers, walls, and agent's position are randomly set according to the configurations of each task.

## C.1 KAREL TASKS

### C.1.1 THE TASK PROMPT OF STAIRCLIMBER

| Purpose | Prompt Text |
|---|---|
| Task Name | STAIRCLIMBERSPARSE |
| Map Description | The map is a 12x12 grid surrounded by walls with stairs formed by walls and a marker is randomly initialized on the stairs as a goal. |
| Initial Position | The agent starts on a random position on the stairs facing east. |
| Task Goal | The goal of the agent is to reach a marker that is also randomly initialized on the stairs. |
| Task Reward | If the agent reaches the marker, the agent receives 1 as an episodic return and 0 otherwise. If the agent moves to an invalid position, i.e. outside the contour of the stairs, the episode terminates with a -1 return. |

### C.1.2 THE TASK PROMPT OF MAZE

| Purpose | Prompt Text |
| --- | --- |
| Task Name | MAZESPARSE |
| Map Description | The map is a complex 8x8 grid surrounded by walls and a random marker is placed on an empty cell as a goal. |
| Initial Position | The agent starts on a random empty cell of the map facing east. |
| Task Goal | The goal of the agent is to reach the goal marker. |
| Task Reward | If the agent reaches the marker, the agent receives 1 as an episodic return and 0 otherwise. |

### C.1.3 THE TASK PROMPT OF FOURCORNERS

| Purpose | Prompt Text |
| --- | --- |
| Task Name | FOURCORNERS |
| Map Description | The map is an empty 12x12 grid surrounded by walls. |
| Initial Position | The agent starts on a random cell on the bottom row of the map facing east. |
| Task Goal | The goal of the agent is to place one marker in each corner of the map. |
| Task Reward | Return is given by the number of corners with one marker divided by 4. |

### C.1.4 THE TASK PROMPT OF TOPOFF

| Purpose | Prompt Text |
| --- | --- |
| Task Name | TOPOFF |
| Map Description | The map is a 12x12 grid surrounded by walls with markers randomly placed on the bottom row of the map. |
| Initial Position | The agent starts on the bottom left cell of the map facing east. |
| Task Goal | The goal of the agent is to place one extra marker on top of every marker on the map. |
| Task Reward | Return is given by the number of markers that have been topped off divided by the total number of markers. Picking up the marker will terminate the episode with a -1 return. |

### C.1.5 THE TASK PROMPT OF HARVESTER

| Purpose | Prompt Text |
| --- | --- |
| Task Name | HARVESTER |
| Map Description | The map is a 8x8 grid surrounded by walls that starts with a marker on each cell. |
| Initial Position | The agent starts on a random cell on the bottom row of the map facing east. |
| Task Goal | The goal of the agent is to pick up every marker on the map. |
| Task Reward | Return is given by the number of picked-up markers divided by the total number of markers. |

### C.1.6 THE TASK PROMPT OF CLEANHOUSE

| Purpose | Prompt Text |
|---|---|
| Task Name | CLEANHOUSE |
| Map Description | The map is a complex 14x22 grid made of many connected rooms and is surrounded by walls. There are ten markers randomly placed adjacent to the walls. |
| Initial Position | The agent starts on a fixed cell facing south. |
| Task Goal | The goal of the agent is to pick up every marker on the map. |
| Task Reward | Return is given by the number of picked-up markers divided by the total number of markers. |

## C.2 KAREL-HARD TASKS

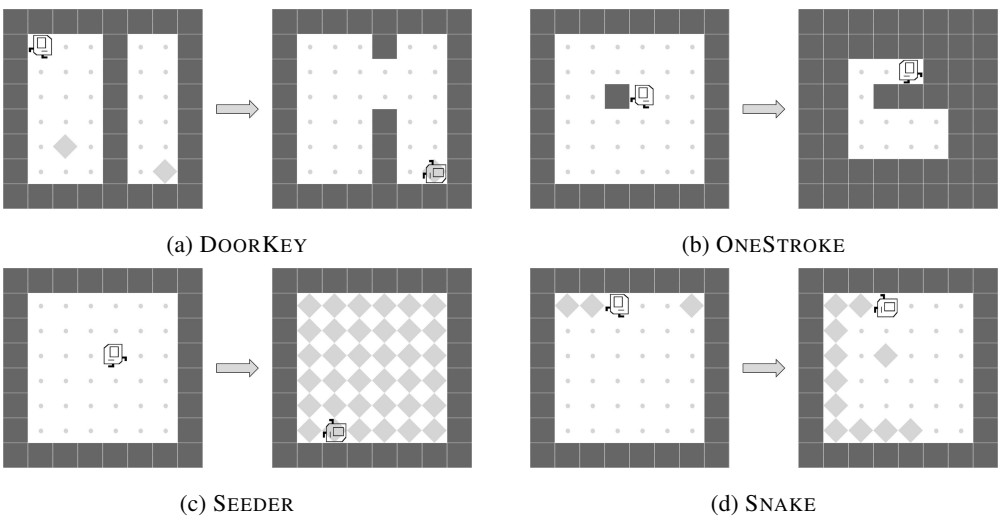

(a) DOORKEY

(b) ONESTROKE

(c) SEEDER

(d) SNAKE

Figure 12: Illustrations of the initial and final state of each task in the Karel-Hard problem set introduced in by Liu et al. (2023). The position of markers, walls, and agent's position are randomly set according to the configurations of each task.

### C.2.1 THE TASK PROMPT OF DOORKEY

| Purpose | Prompt Text |
|---|---|
| Task Name | DOORKEY |
| Map Description | The map is a 8x8 grid surrounded by walls that is vertically split into two chambers. The left chamber is 6x3 grid and the right chamber is 6x2 grid. There is a marker placed randomly on the left chamber as a key, and another marker placed randomly on the right chamber as a goal. |
| Initial Position | The agent starts on a random cell on the left chamber facing east. |
| Task Goal | The goal of the agent is to pick up a marker on the left chamber, which opens a door connecting both chambers. Allow the agent to reach and put a marker on the goal marker. |
| Task Reward | Picking up the first marker yields a 0.5 reward, and putting a marker on the goal marker yields an additional 0.5. |

### C.2.2 THE TASK PROMPT OF ONESTROKE

| Purpose | Prompt Text |
| --- | --- |
| Task Name | ONESTROKE |
| Map Description | The map is given by an empty 8x8 grid surrounded by walls. |
| Initial Position | The agent starts on a random cell of the map facing east. |
| Task Goal | The goal of the agent is to visit every grid cell without repeating. Visited cells become a wall that terminates the episode upon touching. |
| Task Reward | Return is given by the number of visited cells divided by the total number of empty cells in the initial state. |

### C.2.3 THE TASK PROMPT OF SEEDER

| Purpose | Prompt Text |
| --- | --- |
| Task Name | SEEDER |
| Map Description | The map is given by an empty 8x8 grid surrounded by walls.. |
| Initial Position | The agent starts on a random cell of the map facing east. |
| Task Goal | The goal of the agent is to place one marker in every empty cell of the map. |
| Task Reward | Return is given by the number of cells with one marker divided by the total number of empty cells in the initial state. |

### C.2.4 THE TASK PROMPT OF SNAKE

| Purpose | Prompt Text |
| --- | --- |
| Task Name | SNAKE |
| Map Description | The map is given by an empty 8x8 grid surrounded by walls with a marker randomly placed on the map. |
| Initial Position | The agent starts on a random cell of the map facing east. |
| Task Goal | The agent acts like the head of a snake, whose body grows each time a marker is reached. (No need to pick it up.) Every time a marker is reached, the body of the agent grows one marker. The goal of the agent is to touch the marker on the map without colliding with the snake's body, which terminates the episode. Each time the marker is reached, it is placed on a random cell, until 20 markers are reached. |
| Task Reward | Return is given by the number of reached markers divided by 20. |

### C.3 NEW KAREL TASKS

To showcase the extensibility of our proposed LLM-GS framework, we additionally propose two novel new tasks, PATHFOLLOW and WALLAVOIDER. Here we list the task prompts.

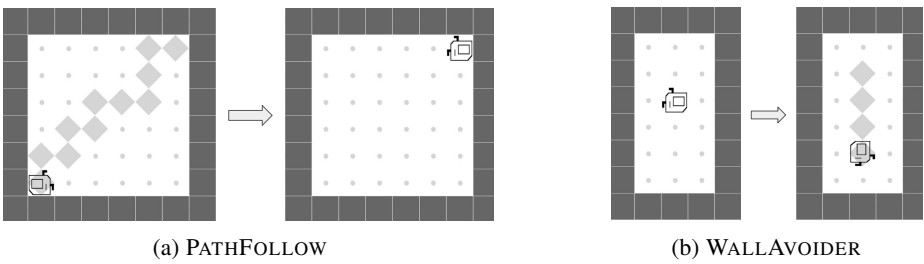

(a) PATHFOLLOW        (b) WALLAVOIDER

Figure 13: Illustrations of the initial and final state of the two new proposed tasks.

### C.3.1 THE TASK PROMPT OF PATHFOLLOW

| Purpose | Prompt Text |
| --- | --- |
| Task Name | PATHFOLLOW |
| Map Description | The map is given by a 8x8 grid surrounded by walls. There is a rugged ascending markers line that starts from the bottom left cell and randomly grows either north or to the east until it reaches the top right cell. Resulting in a rugged markers line connecting the bottom left cell and the top right cell. |
| Initial Position | The agent starts on the bottom left cell of the map facing north. |
| Task Goal | The goal of the agent is to collect every marker on that rugged markers line without leaving the rugged markers line two cells away. |
| Task Reward | Return is given by the number of picked-up markers divided by the total number of markers. Placing any marker or leaving the rugged markers line two cells away will have a negative return as -1.0 and terminate the episode. |

### C.3.2 THE TASK PROMPT OF WALLAVOIDER

| Purpose | Prompt Text |
| --- | --- |
| Task Name | WALLAVOIDER |
| Map Description | The map is given by an empty 8x5 grid surrounded by walls. |
| Initial Position | The agent starts on a random cell of the map facing random directions. |
| Task Goal | The goal of the agent is to place exactly one marker in every interior cell of the map, which refers to the cells that are not adjacent to any wall. |
| Task Reward | Return is given by the number of interior cells with exactly one marker divided by the total number of interior cells. Picking up the marker, putting more than one marker on one cell, or putting any marker on the cell adjacent to any wall will terminate the episode with a -1 return. |

## D AN EXAMPLE PIPELINE OF OUR METHOD

This is our pipeline of getting programs from LLM with the task DOORKEY. Appendix D.1 details the system prompt, Appendix D.2 provides all the information in the task DOORKEY. Appendix D.3 lists the LLM response, and Appendix D.4 gives an example of how we get the Karel DSL program.

### D.1 THE SYSTEM PROMPT

```
You're currently navigating within a Karel environment, which is
    essentially a grid world. In this context, a "world" is
    referred to as a "map." Within this map, there's an entity
```

known as the "agent," capable of movement, changing direction,
  as well as picking up and placing markers on the map.
  Additionally, there are obstacles called "walls" that impede
  the agent's progress; whenever the agent encounters a wall, it
   turns around. Furthermore, there are pre-existing "markers"
  scattered throughout the map at the beginning, though the
  agent has the ability to both pickup and place these markers
  as needed.

Your objective is to generate the appropriate Python program based
   on a given task name and description. This Python program
  will encompass actions enabling the agent to engage with the
  environment, alongside perceptions facilitating the agent's
  recognition of the environment's dynamics.

Here are the available actions for the agent:
move(): Asks the agent to move forward one cell. The agent will
  instead turn left twice if a wall is blocking its way.
turnLeft(): Asks the agent to rotate 90 degrees counterclockwise.
turnRight(): Asks the agent to rotate 90 degrees clockwise.
pickMarker(): Asks the agent to pick up one marker from the
  current cell.
putMarker(): Asks the agent to put down one marker on the current
  cell.

Here are the available perceptions of the agent:
frontIsClear(): Returns True if there is no wall in front of the
  agent.
leftIsClear(): Returns True if there is no wall on the agent's
  left.rightIsClear(): Returns True if there is no wall on the
  agent's right.
markersPresent(): Returns True if there exist markers on the
  current cell.
noMarkersPresent(): Returns True if there is no marker on the
  current cell.

There are some limitations for the Python program:
- do not define other functions besides run()
- do not call other functions
- do not define variables
- do not use True, False, break, continue, return, ==, !=, elif,
  or, and

Python to Karel dsl conversion
1. "def run(): s" to "DEF run m( s m)"
2. "while b: s" to "WHILE c( b c) w( s w)"
3. "if b: s" to "IF c( b c) i( s I)"
4. "if b: s else: s" to "IFELSE c( b c) i( s i) ELSE e( s e)"
5. "for i in range(n): s" to "REPEAT R=n r( s r)"
6. "not h" to "not c( h c)"
7. "frontIsClear()" to "frontIsClear"
8. "leftIsClear()" to "leftIsClear"
9. "rightIsClear()" to "rightIsClear"
10. "markersPresent()" to "markersPresent"
11. "noMarkersPresent()" to "noMarkersPresent"
12. "move()" to "move"
13. "turnLeft()" to "turnLeft"
14. "turnRight()" to "turnRight"
15. "putMarker()" to "putMarker"

```
16. "pickMarker()" to "pickMarker"
```

## D.2 THE USER PROMPT OF THE TASK DOORKEY

```
I'll provide you with the task name and description.

Task name: DOORKEY
Task map: The map is a 8x8 grid surrounded by walls that is
    vertically split into two chambers. The left chamber is 6x3
    grid and the right chamber is 6x2 grid. There is a marker
    placed randomly on the left chamber as a key, and another
    marker placed randomly on the right chamber as a goal.
Task agent position: The agent starts on a random cell on the left
     chamber facing east.
Task goal: The goal of the agent is to pick up a marker on the
    left chamber, which opens a door connecting both chambers.
    Allow the agent to reach and put a marker on the goal marker.
Task return: Picking up the first marker yields a 0.5 reward, and
    putting a marker on the goal marker yields an additional 0.5.

1. Generate 1 simple and short Python program to tackle the task,
    avoid using comments.
2. Convert the Python program to the Karel dsl program.
```

## D.3 THE SAMPLE RESPONSE OF LLM IN THE TASK DOORKEY

```
### Python Program for Task DOORKEY

```python
def run():
    while not markersPresent():
        if frontIsClear():
            move()
        else:
            turnLeft()

    pickMarker()

    turnLeft()

    while not markersPresent():
        if frontIsClear():
            move()
        else:
            turnLeft()

    putMarker()
```

### Karel DSL Conversion

```
DEF run m(
    WHILE not c( markersPresent c) w(
        IF frontIsClear i(
            move i)
        ELSE e(
            turnLeft e) w)
```

```
    pickMarker m
    turnLeft m
    WHILE not c( markersPresent c) w(
        IF frontIsClear i(
            move i)
        ELSE e(
            turnLeft e) w)
    putMarker m)
```
```

### D.4   APPLY POST-PROCESSING AND RETRIEVE THE LLM-INITIALIZED PROGRAMS

For the Python program part in LLM response, we convert the Python program to a DSL program
through the rule-based method and apply post-processing to correct minor mistakes. For the DSL part,
we directly apply post-processing to correct minor mistakes. Appendix D.4.1 presents the original
Python program, the converted DSL, and the post-processed DSL. Appendix D.4.2 shows the original
DSL and the post-processed DSL. All of the post-processing rules are listed in Table 6.

#### D.4.1   PYTHON PROGRAM PART

Listing 1: Original Python program

```python
def run():
    while not markersPresent():
        if frontIsClear():
            move()
        else:
            turnLeft()

    pickMarker()

    turnLeft()

    while not markersPresent():
        if frontIsClear():
            move()
        else:
            turnLeft()

    putMarker()
```

Listing 2: DSL program converted through the rule-based method

```
DEF run m(
    WHILE c( not c( markersPresent c) c) w(
        IFELSE c( frontIsClear c) i(
            move i)
        ELSE e(
            turnLeft e) w)
    pickMarker
    turnLeft
    WHILE c( not c( markersPresent c) c) w(
        IFELSE c( frontIsClear c) i(
            move i)
        ELSE e(
            turnLeft e) w)
    putMarker m)
```

Listing 3: DSL program after post-processing

```
DEF run m(
    WHILE c( noMarkersPresent c) w(
        IFELSE c( frontIsClear c) i(
            move i)
        ELSE e(
            turnLeft e) w)
    pickMarker
    turnLeft
    WHILE c( noMarkersPresent c) w(
        IFELSE c( frontIsClear c) i(
            move i)
        ELSE e(
            turnLeft e) w)
    putMarker m)
```

### D.4.2  DSL PROGRAM PART

Listing 4: Original DSL program

```
DEF run m(
    WHILE not c( markersPresent c) w(
        IF frontIsClear i(
            move i)
        ELSE e(
            turnLeft e) w)
    pickMarker m
    turnLeft m
    WHILE not c( markersPresent c) w(
        IF frontIsClear i(
            move i)
        ELSE e(
            turnLeft e) w)
    putMarker m)
```

Listing 5: DSL program after post-processing

```
DEF run m(
    WHILE c( noMarkersPresent c) w(
        IFELSE c( frontIsClear c) i(
            move i)
        ELSE e(
            turnLeft e) w)
    pickMarker
    turnLeft
    WHILE c( noMarkersPresent c) w(
        IFELSE c( frontIsClear c) i(
            move i)
        ELSE e(
            turnLeft e) w)
    putMarker m)
```

## E  SCHEDULED HILL CLIMBING DETAIL

This Scheduler is designed for improving sample efficiency. $n$ is the number of programs evaluated, and $K_{start}$ and $K_{end}$ are hyper-parameters representing the initial and terminal number of neighbors to search, respectively. $r(n)$ is a sinusoidal function that smoothly increase from $0$ to $1$ over the course of $N$ total programs evaluated in the environment. We provide detailed equation in Equation (2) and Equation (3)

$$log_2 k(n) = (1 - r(n)) \log_2 K_{start} + r(n) \log_2 K_{end}, \tag{2}$$

$$2r(n) = \sin\left[\left(\frac{2 \log n}{\log N} - 1\right) \times \frac{\pi}{2}\right] + 1. \tag{3}$$

### E.1 DESIGN INTUITION

Scheduled HC comprises logarithmic interpolation, sinusoidal schedule, and logarithmic ratio. Our intuition to logarithmic interpolation and logarithmic ratio aims to allocate the most appropriate neighborhood size, $k$, to tasks of varying difficulties. The logarithm of the number of executed programs provides an indication of task difficulty, with the intuition that the optimal $k$ should increase exponentially according to the structure of AST. For the sinusoidal schedule, we prioritize maintaining a stable neighborhood size $k$ during both the early and final stages of the training process.

### E.2 VARIANTS OF SCHEDULED HILL CLIMBING COMPONENTS

We add their linear counterpart, resulting in 2 variants for each component and 8 variants (2x2x2=8) on the whole.

#### E.2.1 INTERPOLATION VARIANTS

- **Logarithmic interpolation**

$$log_2 k(n) = (1 - r(n)) \log_2 K_{start} + r(n) \log_2 K_{end} \tag{4}$$

- **Linear interpolation**

$$k(n) = (1 - r(n)) K_{start} + r(n) K_{end} \tag{5}$$

#### E.2.2 SCHEDULE VARIANTS

- **Sinusoidal schedule**

$$2r(n) = \sin\left[(2 \times ratio - 1) \times \frac{\pi}{2}\right] + 1 \tag{6}$$

- **Linear schedule**

$$r(n) = ratio \tag{7}$$

#### E.2.3 RATIO VARIANTS

- **Logarithmic ratio**

$$\frac{\log n}{\log N} \tag{8}$$

- **Linear ratio**

$$\frac{n}{N} \tag{9}$$

### E.3 ABLATION STUDY RESULTS

We conduct experiments for all 8 variants on DOORKEY and CLEANHOUSE for 8 seeds. The results shown in Figure 14 present no significant differences between all 8 variants.

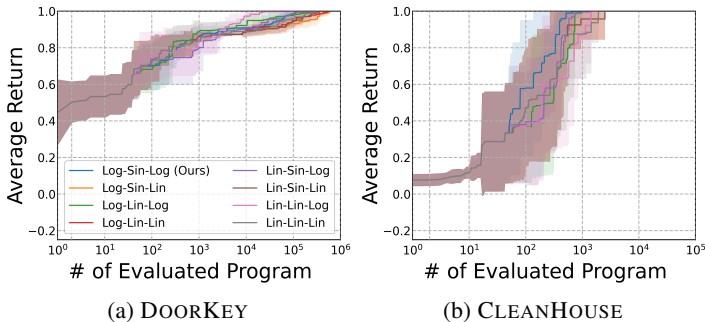

(a) DOORKEY           (b) CLEANHOUSE

Figure 14: **Comparing Scheduled HC variants.** We compare our proposed Scheduled HC with its variants. The label stands for interpolation-schedule-ratio. For example, when using logarithmic interpolation, sinusoidal schedule, and logarithmic ratios, our label is *Log-Sin-Log*. The results indicate no significant differences between all Scheduled HC variants.

## F  BASELINES AND THEIR HYPERPARAMETERS

We compare our framework to four baselines, LEAPS (Trivedi et al., 2021), HPRL (Liu et al., 2023), CEBS (Carvalho et al., 2024), and HC (Carvalho et al., 2024). LEAPS, HPRL, and CEBS search programs in the latent spaces, while HC searches programs in the programmatic space. We first introduce the concept of latent space in Appendix F.1, then dive into the details of each baseline in the following sections. The pseudo-code of CEBS is described in Algorithm 1, and the pseudo-code of HC is described in Algorithm 2.

### F.1  LATENT SPACE

In addition to the programmatic space, another choice is to search in a learned *latent space* (Trivedi et al., 2021; Liu et al., 2023), constructed by training a variational autoencoder on randomly generated DSL programs. To sample a neighbor program, a small noise is added to the latent embedding of the current program candidate, and the decoder can generate a program from the noise-corrupted embedding.

### F.2  LEAPS

LEAPS utilizes the Cross-entropy method (CEM) (Rubinstein and Kroese, 2004) as its search strategy. It first generates $k$ neighborhood programs from its latent space and evaluates their episodic return on the Karel environment (Trivedi et al., 2021). Next, if the average return over the top-$E$ neighborhood programs is higher than the best-seen average return, it continues the process on the mean latent embedding of the top-$E$ neighborhood programs. When the average return no longer increases, the program with the highest episodic return is returned.

The hyperparameters of the LEAPS baseline are from LEAPS (Trivedi et al., 2021) and HPRL (Liu et al., 2023). We downloaded the pre-trained weights of LEAPS and used the parameters for searching programs. We use the re-implementation of LEAPS from Carvalho et al. (2024), thus we do not have hyperparameters of exponential decay. Also, the re-implementation samples candidate from a fixed normal distribution which is hyperparameter in LEAPS (Trivedi et al., 2021). In Table 7, $k$ is the neighborhood size and $E$ is the candidate, $\sigma$ is the noise scale. In LEAPS and HPRL, they use an elite ratio to represent the candidate programs. For better understanding, we round $E$ to be an integer.

Table 7: LEAPS hyperparameter settings on all KAREL tasks.

| TASK NAME | $k$ | $\sigma$ | $E$ |
|---|---|---|---|
| STAIRCLIMBER | 32 | 0.25 | 2 |
| FOURCORNERS | 64 | 0.5 | 13 |
| TOPOFF | 64 | 0.25 | 3 |
| MAZE | 16 | 0.1 | 2 |
| CLEANHOUSE | 32 | 0.25 | 2 |
| HARVESTER | 32 | 0.5 | 3 |
| DOORKEY | 32 | 0.25 | 3 |
| ONESTROKE | 64 | 0.5 | 3 |
| SEEDER | 32 | 0.25 | 3 |
| SNAKE | 32 | 0.25 | 6 |

### F.3 HIERARCHICAL PROGRAMMATIC REINFORCEMENT LEARNING (HPRL)

HPRL (Liu et al., 2023) aims to improve LEAPS by composing several programs to represent more complex behaviors. Given a learned latent space, meta-policy learns to predict a sequence of actions, *i.e.* programs, by optimizing the return obtained by executing the composed programs using reinforcement learning. HPRL considers the discount factor $\gamma$ in the meta-MDP; as a result, the evaluation method is not the same as our problem formulation. To ensure a fair comparison, we record one million programs explored by meta-policy in the training stage. In HRPL, the authors use the training step as a hyperparameter. One training step is one step of meta-policy, and the maximum episode length is set to 5 as the original setting in the HPRL paper. In Table 8, we list the hyperparameters we modified in the HPRL training script. All other hyperparameters remain the same as described in HPRL.

Table 8: HRPL hyperparameter settings on all KAREL tasks.

| TASK NAME | training steps | height | weight |
|---|---|---|---|
| STAIRCLIMBER | 50K | 12 | 12 |
| FOURCORNERS | 500K | 12 | 12 |
| TOPOFF | 5M | 12 | 12 |
| MAZE | 50K | 8 | 8 |
| CLEANHOUSE | 500K | 12 | 22 |
| HARVESTER | 5M | 8 | 8 |
| DOORKEY | 5M | 8 | 8 |
| ONESTROKE | 5M | 8 | 8 |
| SEEDER | 5M | 8 | 8 |
| SNAKE | 5M | 8 | 8 |

### F.4 CROSS-ENTROPY BEAM SEARCH (CEBS)

CEBS (Carvalho et al., 2024) extends CEM to maintain a set of $E$ candidate programs to perform beam search. Unlike CEM which only maintains one candidate program. CEBS maintains a set of $E$ candidate programs, to perform beam search. In other words, CEBS searches all the neighborhoods of the top-$E$ programs.

It also utilizes the pre-trained VAE weight from LEAPS (Trivedi et al., 2021) to search the program in the latent space. All the hyperparameter follows the original CEBS with neighborhood $K$ equal to 64, candidate $E$ equal to 16, and noise ratio $\sigma$ equal to 0.25 for all ten Karel tasks. The pseudo-code is described in Algorithm 1.

## F.5 Hill climbing (HC)

HC (Carvalho et al., 2024) is a state-of-the-art algorithm solving Programmatic Reinforcement Learning tasks. All the hyperparameter follows the original HC with neighborhood $K$ equal to 250 for all ten Karel tasks. The pseudo-code is described in Algorithm 2.

---

**Algorithm 1** Cross-entropy beam search algorithm  (Carvalho et al., 2024)

---

**Require:** $k$, number of neighborhood; $E$ number of top candidate; $T$, the task; VAE, the program encoder and decoder.
**Ensure:** $\rho^*$ the highest averaged return over 32 task variants.
1: $z \sim N(0, \mathbf{I})$
2: $\rho \leftarrow \text{VAE.decode}(z)$
3: $steps \leftarrow 0$
4: $Return \leftarrow \text{evaluate}(\rho, T)$
5: $\rho^* \leftarrow \rho$
6: $Mean \leftarrow -\infty$
7: $P \leftarrow \text{get-neighbor}(\rho, k)$
8: **while** $steps < 1000000$ **do**
9:    $Candidates \leftarrow [\,]$
10:    **for** each $\rho_{\text{new}}$ in $P$ **do**
11:       $r \leftarrow \text{evaluate}(\rho_{\text{new}}, T)$
12:       $Candidates.\text{append}(r, \rho_{\text{new}})$
13:       **if** $r > Return$ **then**
14:          $\rho^* \leftarrow \rho_{\text{new}}$
15:          $Return \leftarrow r$
16:       **end if**
17:    **end for**
18:    $Elites \leftarrow \text{Top-E}(Candidates)$
19:    **if** $Mean > \text{get-mean}(Elites)$ **then**
20:       **break**
21:    **end if**
22:    **if** $Return = 1$ **then**
23:       **break**
24:    **end if**
25:    $Mean \leftarrow \text{get-mean}(Elites)$
26:    $P \leftarrow [\,]$
27:    **for**   each $\rho_{\text{new}}$ in $Elites$ **do**
28:       $P.\text{extend}(\text{get-neighbor}(\rho_{\text{new}}, k/E))$
29:    **end for**
30:    $steps \leftarrow steps + k$
31: **end while**

---

## G  Data Leakage

While our proposed framework shows significant improvement over the baselines, some may question if this improvement only comes from the LLM "memorizing" all the answers. Indeed, some ground truth solutions have been documented in the previous literature (Trivedi et al., 2021; Liu et al., 2023; Carvalho et al., 2024), thus making data leakage a potential concern. Still, we would like to rule out this possibility from three aspects: the timeline of previous works, the LLMs' understanding of the Karel tasks, and the innovation of two novel tasks. Thus proving our framework utilizes the LLMs' understanding and reasoning ability not just its internal knowledge.

**Timeline.** For the Karel-Hard problem set introduced by Liu et al. (2023) in July 2023, the optimal programs were not included in Liu et al. (2023) since their proposed method could only partially solve the tasks. Carvalho et al. (2024), made public in 2024, are the first to provide optimal programs for the Karel-Hard tasks. Hence, our LLM (gpt-4-turbo-2024-04-09 with knowledge up to December 2023) could not access the optimal Karel-Hard tasks programs.

---

**Algorithm 2** Hill climbing algorithm  (Carvalho et al., 2024)

---

```
Require: T, the task; k, number of neighborhood.
Ensure: ρ*, the highest averaged return over 32 task variants.
 1: Initialize ρ with a random solution
 2: Return ← evaluate(ρ, T)
 3: ρ* ← ρ
 4: improved ← True
 5: steps ← 0
 6: while improved and steps < n do
 7:    improved ← False
 8:    Neighbors ← get-neighbor(ρ,k)
 9:    for each ρ_new in Neighbors do
10:       r_new ← evaluate(ρ_new, T)
11:       if r_new > Return then
12:          ρ* ← ρ_new
13:          Return ← r_new
14:          improved ← True
15:          break
16:       end if
17:       steps ← steps + 1
18:    end for
19:    ρ ← ρ*
20: end while
```

---

> The Karel environment is used to test Programmatic Reinforcement Learning (PRL) algorithms and includes a variety of tasks. There are six tasks in the Karel test set and four more challenging tasks in the Karel-hard test set. These tasks are utilized in the research papers "Learning to Synthesize Programs as Interpretable and Generalizable Policies" and "Hierarchical Programmatic Reinforcement Learning via Learning to Compose Programs."
> Can you list all ten tasks and their task descriptions based on what you have learned?

Figure 15: The prompt that asks GPT-4 about the Karel PRL tasks

**Does the LLM know the Karel tasks?.** To verify if the LLM knows the Karel tasks used in our work, we prompt GPT-4 by providing the paper names of LEAPS (Trivedi et al., 2021) and HPRL (Liu et al., 2023) and ask GPT-4 to list all the names and descriptions of the Karel tasks used in these works. Among all the responses generated by GPT-4, the only task closely resembling the Karel tasks used in our work is "Maze Solving." However, all other tasks described in the GPT-4's response are drastically distinct from the ones we used in our work. The full prompt and response are listed in Figure 15 and Figure 16.

**Two novel tasks.** We additionally design two novel tasks that have not been used in any existing work, PATHFOLLOW, and WALLAVOIDER, as illustrated in Figure 13. The details of these two novel tasks are in Appendix C.3. The results in Figure 8 show that our framework can extend to new tasks that are impossible to leak to LLMs.

## H    AGGREGATED PERFORMANCE ACROSS ALL TASKS

To clearly examine the performance across all the tasks, we include a plot that aggregates the performance of all main results (Figure 4) in Figure 17. Our proposed LLM-GS framework significantly outperforms the existing methods.

Sure! The Karel environment, based on the educational programming language designed to introduce students to the basics of programming, provides programmatic challenges that both regular and hierarchical reinforcement learning algorithms can address. Here is a summary of the tasks typically included in both the Karel test set and the Karel-hard set based on common challenges noted in similar environments and research contexts:

KAREL TEST SET TASKS

1. **Move to Wall**: The task is for the agent (Karel) to move forward until it encounters a wall.

2. **Turn Right**: Teach Karel to turn right using a sequence of commands (since Karel inherently only has a command to turn left).

3. **Collect Marker**: Karel needs to navigate towards and collect a marker from the grid.

4. **Avoid Hurdles**: Karel must move from one point to another while avoiding hurdles placed in its path.

5. **Put Marker and Move**: This task combines placing a marker in a specified location and then moving to a different location.

6. **Maze Solving**: Karel needs to find a path from one point to another in a maze-like structure.

KAREL-HARD TEST SET TASKS

1. **Multiple Markers**: Karel might need to collect or place multiple markers in various locations, requiring more complex navigation and memory usage.

2. **Dynamic Obstacles**: The environments where obstacles may change location or appear suddenly, requiring Karel to adapt its path dynamically.

3. **Conditional Logic Usage**: Tasks where Karel must use sensors to make decisions based on the conditions of its environment (e.g., If front is clear, move forward; otherwise, turn left).

4. **Complex Maze**: More intricate and challenging mazes that require advanced pathfinding algorithms and possibly backtracking strategies.

While these are typical tasks deduced from the context of using Karel for testing PRL algorithms, please note that specific details or task names might vary based on the particular implementation or the focuses of different research papers or setups within the "Karel" environment. Always refer back to the specific research documents or source materials for the most accurate descriptions and objectives of these tasks.

Figure 16: The GPT-4 response about Karel PRL tasks

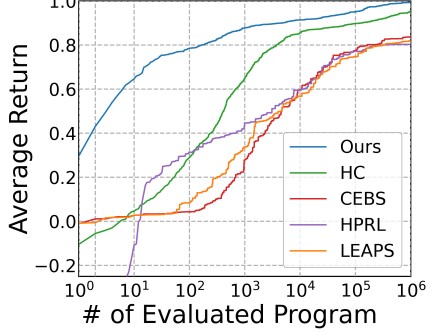

Figure 17: **Aggregated performance across all the tasks.** The aggregate performance across all ten tasks. Our proposed LLM-GS framework surpasses all existing methods by a large margin.

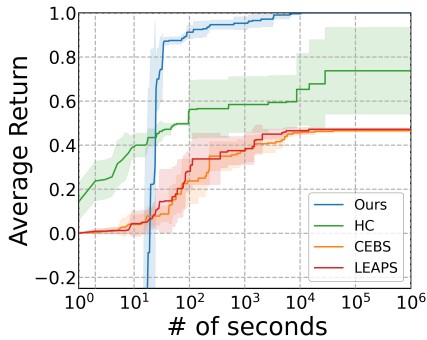

Figure 18: **Wall-clock time evaluation on the DOORKEY task.** We evaluate the wall-clock time performance on the DOORKEY task, with the x-axis representing the number of seconds on a logarithmic scale. The plot also includes the time LLM takes to generate the programs. The results demonstrate the superiority of our proposed LLM-GS framework.

## I WALL-CLOCK TIME EVALUATION ON TASK DOORKEY

We provide a wall-clock time evaluation (real-time elapsed) for the DOORKEY task in Figure 18, with the x-axis showing the number of seconds on a log scale. It also accounts for the time LLM takes to generate the programs. The result highlights the superiority of our proposed LLM-GS framework.

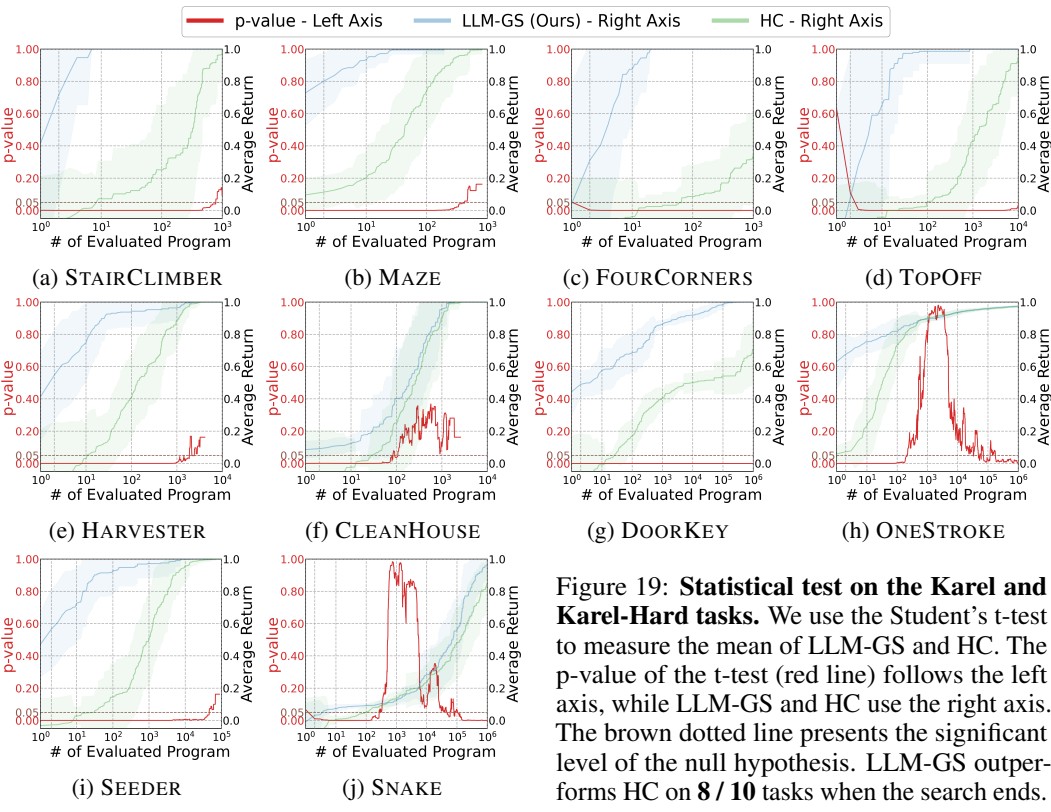

Figure 19: **Statistical test on the Karel and Karel-Hard tasks.** We use the Student's t-test to measure the mean of LLM-GS and HC. The p-value of the t-test (red line) follows the left axis, while LLM-GS and HC use the right axis. The brown dotted line presents the significant level of the null hypothesis. LLM-GS outperforms HC on **8 / 10** tasks when the search ends.

## J STATISTICAL TEST BETWEEN LLM-GS AND HC

Since the confidence interval of LLM-GS overlaps with that of HC in some tasks, we conduct hypothesis testing between these two methods for better comparison. The null hypothesis $H_0$ is that the mean of LLM-GS is greater than HC. We use Student's t-test (STUDENT, 1908) to test the

distribution of LLM-GS and HC at the same number of evaluated programs. Figure 19 shows the p-value from the t-test across the evaluated program on the Karel and Karel-hard tasks. We adjust the primary y-axis (left) and the secondary y-axis (right) to have the same scale. The brown dotted line is 0.05. If the red lines (p-value) are under the brown dotted line, LLM-GS statistically outperforms HC. If the red lines are above the brown dotted line, LLM-GS fails to surpass HC.

The result shows that our method significantly outperforms HC on STAIRCLIMBER, MAZE, FOUR-CORNERS, DOORKEY, and SEEDER. On task TOPOFF, LLM initialization can not outperform HC in the first few attempts but has superior efficiency when the search ends. On the task HARVESTER, LLM-GS can surpass HC in less than a thousand programs, but cannot find a difference at the end of the search. On the task CLEANHOUSE, LLM-GS cannot outperform HC except for the LLM initialization. On the task ONESTROKE and SNAKE, LLM-GS eventually surpasses HC when the search process ends. The LLM initialization (32 is at the middle point of 10 and 100) surpasses HC on all tasks. On the other hand, LLM-GS outperforms HC on **8 / 10** tasks (except for HARVESTER and CLEANHOUSE) when the search ends.

## K  LLM PROMPTS FOR ABLATIONS AND REVISION

**Ablation prompts for program generation methods.** We conduct an ablation study in Section 5.3 to justify that our LLM-generating program method performs best in acceptance rate and best return. Here we list the complete prompts. There are three approaches to generating DSL programs: Pythonic-DSL, Python, and DSL. There are two types of prompt, system prompt and user prompt, in all approaches. In the system prompt, both the Python and the Pythonic-DSL approaches contain limitations of Python usage. The DSL approach contains the grammar of the Karel, and the Pythonic-DSL contains the paired Python-like and Karel production rules. All of the system prompts contain the environment physics, action, and perception. On the other hand, the user prompt contains five placeholders for task name, map description, initial position, task goal, and task reward. In Appendix C, all of the task-dependent information can be filled in the placeholders. Appendix D.2 shows the full Pythonic-DSL prompt with the task DOORKEY filled in the placeholders. The Pythonic-DSL prompts are in Appendix K.1, and the Python prompts are in Appendix K.2, and the DSL prompts are in Appendix K.3.

**Prompts for LLM revision.** We list all of the user prompts in the experiment of LLM revision, the system prompt is the same as the one in Appendix K.1. We implement four approaches to revising the program: Regenerate, Regenerate with reward, Agent execution trace, and Agent and program execution trace. We ask the LLM to regenerate with all generated programs in the last round without repetition, and the Regenerate prompt is in Appendix K.4. We ask the LLM to regenerate with the program relating to reward, and the revision prompt of the Regenerate with reward is in Appendix K.5. We utilize the execution trace of the Karel agent in the grid world, and the revision prompt of the Agent execution trace is in Appendix K.6. We provided the action/perception call and executing line, and the revision prompt of the Agent and program execution trace is in Appendix K.7. The program Listing 1 is the revision target of our method revision method in Appendix K.6 and Appendix K.7. Figure 20 is the trajectories of the program we used in method Appendix K.6 and Appendix K.7. In the example, the origin program has an average return of 0.5. LLM revision with agent trace can reach 0.640625, while LLM revision with both agent and program traces can reach a result of 0.8125.

### K.1  PYTHONIC-DSL

#### K.1.1  SYSTEM PROMPT

```
You're currently navigating within a Karel environment, which is
   essentially a grid world. In this context, a "world" is
   referred to as a "map." Within this map, there's an entity
   known as the "agent," capable of movement, changing direction,
    as well as picking up and placing markers on the map.
   Additionally, there are obstacles called "walls" that impede
   the agent's progress; whenever the agent encounters a wall, it
    turns around. Furthermore, there are pre-existing "markers"
   scattered throughout the map at the beginning, though the
```

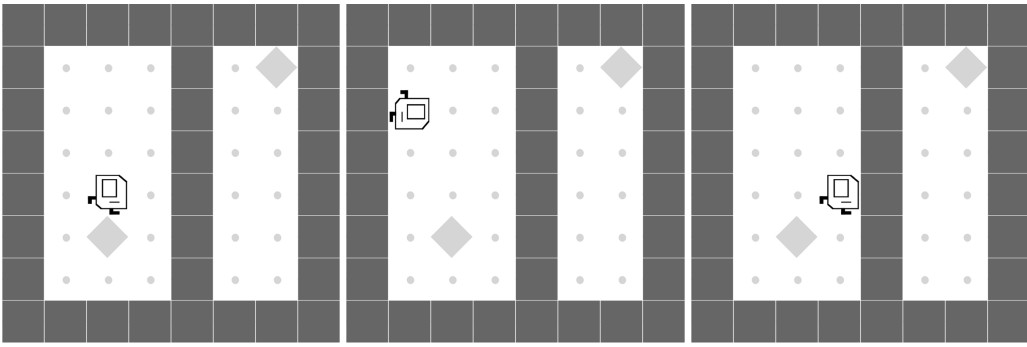

(a) This is the initial state of one of DOORKEY the task variants.

(b) This is the last step of the trajectories in the prompts.

(c) The program ends at step 47 and no reward is granted.

Figure 20: This is the demonstration of the task DOORKEY. The Karel agent can only reach the surrounding grids at the left chamber in the trajectory.

**Listing 1** This is the program for LLM revision.

```
def run():
    while noMarkersPresent():
        if frontIsClear():
            move()
        else:
            turnLeft()
            if frontIsClear():
                move()
                turnRight()
    pickMarker()
    while noMarkersPresent():
        if frontIsClear():
            move()
        else:
            turnLeft()
    putMarker()
```

**Listing 2** The revision results of Appendix K.6.

```
    while noMarkersPresent():
        if frontIsClear():
            move()
        else:
            turnLeft()
    pickMarker()
    while frontIsClear():
        move()
    turnRight()
    while noMarkersPresent():
        if frontIsClear():
            move()
        else:
            turnLeft()
            if frontIsClear():
                move()
                turnRight()
    putMarker()
```

---

**Listing 3** The revision results of Appendix K.7.

```python
    while noMarkersPresent():
        if frontIsClear():
            move()
        else:
            turnLeft()
    pickMarker()
    while noMarkersPresent():
        if rightIsClear():
            turnRight()
            move()
        else:
            if frontIsClear():
                move()
            else:
                turnLeft()
    putMarker()
```

---

```
    agent has the ability to both pickup and place these markers
    as needed.

Your objective is to generate the appropriate Python program based
    on a given task name and description. This Python program
    will encompass actions enabling the agent to engage with the
    environment, alongside perceptions facilitating the agent's
    recognition of the environment's dynamics.

Here are the available actions for the agent:
move(): Asks the agent to move forward one cell. The agent will
    instead turn left twice if a wall is blocking its way.
turnLeft(): Asks the agent to rotate 90 degrees counterclockwise.
turnRight(): Asks the agent to rotate 90 degrees clockwise.
pickMarker(): Asks the agent to pick up one marker from the
    current cell.
putMarker(): Asks the agent to put down one marker on the current
    cell.>

Here are the available perceptions of the agent:
frontIsClear(): Returns True if there is no wall in front of the
    agent.
leftIsClear(): Returns True if there is no wall on the agent's
    left.
rightIsClear(): Returns True if there is no wall on the agent's
    right.
markersPresent(): Returns True if there exist markers on the
    current cell.
noMarkersPresent(): Returns True if there is no marker on the
    current cell.

There are some limitations for the Python program:
- do not define other functions besides run()
- do not call other functions
- do not define variables
- do not use True, False, break, continue, return, ==, !=, elif,
    or, and

Python to Karel dsl conversion
```

```
1.  "def run(): s" to "DEF run m( s m)"
2.  "while b: s" to "WHILE c( b c) w( s w)"
3.  "if b: s" to "IF c( b c) i( s i)"
4.  "if b: s else: s" to "IFELSE c( b c) i( s i) ELSE e( s e)"
5.  "for i in range(n): s" to "REPEAT R=n r( s r)"
6.  "not h" to "not c( h c)"
7.  "frontIsClear()" to "frontIsClear"
8.  "leftIsClear()" to "leftIsClear"
9.  "rightIsClear()" to "rightIsClear"
10. "markersPresent()" to "markersPresent"
11. "noMarkersPresent()" to "noMarkersPresent"
12. "move()" to "move"
13. "turnLeft()" to "turnLeft"
14. "turnRight()" to "turnRight"
15. "putMarker()" to "putMarker"
16. "pickMarker()" to "pickMarker"
```

### K.1.2 USER PROMPT

```
I'll provide you with the task name and description.

Task name: <<Task Name>>
Task map: <<Map Description>>
Task agent position: <<Initial Position>>
Task goal: <<Task Goal>>
Task return: <<Task Reward>>

1. Generate 1 simple and short Python program to tackle the task,
   avoid using comments.
2. Convert the Python program to the Karel dsl program.
```

## K.2 PYTHON

### K.2.1 SYSTEM PROMPT

```
You're currently navigating within a Karel environment, which is
   essentially a grid world. In this context, a "world" is
   referred to as a "map." Within this map, there's an entity
   known as the "agent," capable of movement, changing direction,
    as well as picking up and placing markers on the map.
   Additionally, there are obstacles called "walls" that impede
   the agent's progress; whenever the agent encounters a wall, it
    turns around. Furthermore, there are pre-existing "markers"
   scattered throughout the map at the beginning, though the
   agent has the ability to both pickup and place these markers
   as needed.

Your objective is to generate the appropriate Python program based
    on a given task name and description. This Python program
   will encompass actions enabling the agent to engage with the
   environment, alongside perceptions facilitating the agent's
   recognition of the environment's dynamics.

Here are the available actions for the agent:
move(): Asks the agent to move forward one cell. The agent will
   instead turn left twice if a wall is blocking its way.
turnLeft(): Asks the agent to rotate 90 degrees counterclockwise.
turnRight(): Asks the agent to rotate 90 degrees clockwise.
```

```
pickMarker(): Asks the agent to pick up one marker from the
    current cell.
putMarker(): Asks the agent to put down one marker on the current
    cell.

Here are the available perceptions of the agent:
frontIsClear(): Returns True if there is no wall in front of the
    agent.
leftIsClear(): Returns True if there is no wall on the agent's
    left.
rightIsClear(): Returns True if there is no wall on the agent's
    right.
markersPresent(): Returns True if there exist markers on the
    current cell.
noMarkersPresent(): Returns True if there is no marker on the
    current cell.

There are some limitations for the Python program:
- do not define other functions besides run()
- do not call other functions
- do not define variables
- do not use True, False, break, continue, return, ==, !=, elif,
    or, and
```

### K.2.2 USER PROMPT

```
I'll provide you with the task name and description.

Task name: <<Task Name>>
Task map: <<Map Description>>
Task agent position: <<Initial Position>>
Task goal: <<Task Goal>>
Task return: <<Task Reward>>

1. Generate 1 simple and short Python program to tackle the task,
    avoid using comments.
```

### K.3 DSL

### K.3.1 SYSTEM PROMPT

```
You're currently navigating within a Karel environment, which is
    essentially a grid world. In this context, a "world" is
    referred to as a "map." Within this map, there's an entity
    known as the "agent," capable of movement, changing direction,
     as well as picking up and placing markers on the map.
    Additionally, there are obstacles called "walls" that impede
    the agent's progress; whenever the agent encounters a wall, it
     turns around. Furthermore, there are pre-existing "markers"
    scattered throughout the map at the beginning, though the
    agent has the ability to both pickup and place these markers
    as needed.

Your objective is to generate the appropriate Karel dsl program
    based on a given task name and description. This Karel dsl
    program will encompass actions enabling the agent to engage
    with the environment, alongside perceptions facilitating the
    agent's recognition of the environment's dynamics.
```

```
Here are the available actions for the agent:
move: Asks the agent to move forward one cell. The agent will
    instead turn left twice if a wall is blocking its way.
turnLeft: Asks the agent to rotate 90 degrees counterclockwise.
turnRight: Asks the agent to rotate 90 degrees clockwise.
pickMarker: Asks the agent to pick up one marker from the current
    cell.
putMarker: Asks the agent to put down one marker on the current
    cell.

Here are the available perceptions of the agent:
frontIsClear: Returns True if there is no wall in front of the
    agent.
leftIsClear: Returns True if there is no wall on the agent's left.
rightIsClear: Returns True if there is no wall on the agent's
    right.
markersPresent: Returns True if there exist markers on the current
     cell.
noMarkersPresent: Returns True if there is no marker on the
    current cell.

This is the production role of the domain-specific language of the
     Karel environment.
Program p := DEF run m( s m)
Statement s := WHILE c( b c) w( s w) | IF c( b c) i( s i) | IFELSE
    c( b c) i( s i) ELSE e( s e) | REPEAT R=n r( s r) | s s | a
Condition b := h | not c( h c)
Number n := 0, 1, 2, 3, 4, 5, 6, 7, 8, 9, 10, 11, 12, 13, 14, 15,
    16, 17, 18, 19
Perception h := frontIsClear | leftIsClear | rightIsClear |
    markersPresent | noMarkersPresent
Action a := move | turnLeft | turnRight | putMarker | pickMarker
```

### K.3.2 USER PROMPT

```
I'll provide you with the task name and description.

Task name: <<Task Name>>
Task map: <<Map Description>>
Task agent position: <<Initial Position>>
Task goal: <<Task Goal>>
Task return: <<Task Reward>>

1. Generate 1 simple and short Karel dsl program to tackle the
    task, avoid using comments.
```

### K.4 REGENERATE

```
I'll provide you with the task name, task description, and the
    programs you generated last time.

Task name: DOORKEY
Task map: The map is a 8x8 grid surrounded by walls that is
    vertically split into two chambers. The left chamber is 6x3
    grid and the right chamber is 6x2 grid. There is a marker
    placed randomly on the left chamber as a key, and another
    marker placed randomly on the right chamber as a goal.
```

```
Task agent position: The agent starts on a random cell on the left
    chamber facing east.
Task goal: The goal of the agent is to pick up a marker on the
    left chamber, which opens a door connecting both chambers.
    Allow the agent to reach and put a marker on the goal marker.
Task return: Picking up the first marker yields a 0.5 reward, and
    putting a marker on the goal marker yields an additional 0.5.

These are the programs you generated last time, all of these
    programs cannot yield perfect performance.

Program 1:
def run():
    while noMarkersPresent():
        if frontIsClear():
            move()
        else:
            turnLeft()
    pickMarker()
    while noMarkersPresent():
        if frontIsClear():
            move()
        else:
            turnLeft()
    if markersPresent():
        putMarker()

###23 programs are truncated.###

Program 25:
def run():
    while noMarkersPresent():
        if frontIsClear():
            move()
        else:
            turnLeft()
    pickMarker()
    for i in range(19):
        if frontIsClear():
            move()
        else:
            turnRight()

1. Generate a Python program that is not identical to any of the
    previous programs to tackle the task, and avoid using comments
    .
2. Convert the Python program to the Karel dsl program.
```

## K.5 REGENERATE WITH REWARD

```
I'll provide you with the task name, task description, and the
    programs rewards pairs sorted by their evaluation rewards from
    32 task variants.

Task name: DOORKEY
Task map: The map is a 8x8 grid surrounded by walls that is
    vertically split into two chambers. The left chamber is 6x3
```

```
   grid and the right chamber is 6x2 grid. There is a marker
   placed randomly on the left chamber as a key, and another
   marker placed randomly on the right chamber as a goal.
Task agent position: The agent starts on a random cell on the left
    chamber facing east.
Task goal: The goal of the agent is to pick up a marker on the
   left chamber, which opens a door connecting both chambers.
   Allow the agent to reach and put a marker on the goal marker.
Task return: Picking up the first marker yields a 0.5 reward, and
   putting a marker on the goal marker yields an additional 0.5.

Program reward pairs sorted by their evaluation rewards:

Program 1:
def run():
    while noMarkersPresent():
        if frontIsClear():
            move()
        else:
            turnLeft()
            if frontIsClear():
                move()
                turnRight()
    pickMarker()
    while noMarkersPresent():
        if frontIsClear():
            move()
        else:
            turnLeft()
    putMarker()

reward:
0.5

###23 programs are truncated.###

Program 25:
def run():
    for i in range(19):
        if markersPresent():
            pickMarker()
        while not frontIsClear():
            turnLeft()
        move()
        if frontIsClear():
            move()
        if markersPresent():
            putMarker()

reward:
-0.5

1. Depending on this information, examine the program pattern that
    the highest score programs process, but the lowest score
   programs do not.
2. Generate 1 simple and short Python program according to the
   pattern to tackle the task, avoid using comment.
3. Convert the Python program to the Karel dsl program.
```

## K.6 AGENT EXECUTION TRACE

```
I'll provide you with the code you developed previously, with the
    goal of refining it. To guide your revision, you'll receive
    the specific task name and a description. Since there are 32
    different versions of the task that share the same objective
    but differ by random seeds, I will identify the specific
    variant where the performance of the program is most lacking.
    Additionally, you'll get the initial state of the task, the
    code, and a detailed trajectory demonstrating how the code
    operates within this particular scenario. This trajectory will
     detail each action step-by-step and show a localized snapshot
     of the environment (a 3x3 area centered on the agent) during
    execution. Rewards received by the agent will also be shown
    during these steps.

Task name: DOORKEY
Task map: The map is a 8x8 grid surrounded by walls that is
    vertically split into two chambers. The left chamber is 6x3
    grid and the right chamber is 6x2 grid. There is a marker
    placed randomly on the left chamber as a key, and another
    marker placed randomly on the right chamber as a goal.
Task agent position: The agent starts on a random cell on the left
     chamber facing east.
Task goal: The goal of the agent is to pick up a marker on the
    left chamber, which opens a door connecting both chambers.
    Allow the agent to reach and put a marker on the goal marker.
Task return: Picking up the first marker yields a 0.5 reward, and
    putting a marker on the goal marker yields an additional 0.5.

Initial state:
Wall(0, 0) ;    Wall(0, 1) ;    Wall(0, 2) ;    Wall(0, 3) ;
    Wall(0, 4) ;    Wall(0, 5) ;    Wall(0, 6) ;    Wall(0, 7) ;
Wall(1, 0) ;    Empty(1, 1) ;   Empty(1, 2) ;   Empty(1, 3) ;
    Wall(1, 4) ;    Empty(1, 5) ;   Marker(1, 6, quantity=1) ;
        Wall(1, 7) ;
Wall(2, 0) ;    Empty(2, 1) ;   Empty(2, 2) ;   Empty(2, 3) ;
    Wall(2, 4) ;    Empty(2, 5) ;   Empty(2, 6) ;   Wall(2, 7) ;
Wall(3, 0) ;    Empty(3, 1) ;   Empty(3, 2) ;   Empty(3, 3) ;
    Wall(3, 4) ;    Empty(3, 5) ;   Empty(3, 6) ;   Wall(3, 7) ;
Wall(4, 0) ;    Empty(4, 1) ;   Agent(4, 2, direction=(0, 1)) ;
    Empty(4, 3) ;   Wall(4, 4) ;    Empty(4, 5) ;   Empty(4, 6) ;
        Wall(4, 7) ;
Wall(5, 0) ;    Empty(5, 1) ;   Marker(5, 2, quantity=1) ;
    Empty(5, 3) ;   Wall(5, 4) ;    Empty(5, 5) ;   Empty(5, 6) ;
        Wall(5, 7) ;
Wall(6, 0) ;    Empty(6, 1) ;   Empty(6, 2) ;   Empty(6, 3) ;
    Wall(6, 4) ;    Empty(6, 5) ;   Empty(6, 6) ;   Wall(6, 7) ;
Wall(7, 0) ;    Wall(7, 1) ;    Wall(7, 2) ;    Wall(7, 3) ;
    Wall(7, 4) ;    Wall(7, 5) ;    Wall(7, 6) ;    Wall(7, 7) ;

Program:
def run():
    while noMarkersPresent():
        if frontIsClear():
            move()
        else:
            turnLeft()
            if frontIsClear():
```

```
                move()
                turnRight()
        pickMarker()
        while noMarkersPresent():
            if frontIsClear():
                move()
            else:
                turnLeft()
        putMarker()
```

The average reward on 32 task variants is:
0.5

Trajectory:

Step 1:
Agent performs a perception: noMarkersPresent. The result is True.
Partial state:
Empty(3, 1) ;    Empty(3, 2) ;    Empty(3, 3) ;
Empty(4, 1) ;    Agent(4, 2, direction=(0, 1)) ; Empty(4, 3) ;
Empty(5, 1) ;    Marker(5, 2, quantity=1) ;       Empty(5, 3) ;

Step 2:
Agent performs a perception: frontIsClear. The result is True.
Partial state:
Empty(3, 1) ;    Empty(3, 2) ;    Empty(3, 3) ;
Empty(4, 1) ;    Agent(4, 2, direction=(0, 1)) ; Empty(4, 3) ;
Empty(5, 1) ;    Marker(5, 2, quantity=1) ;       Empty(5, 3) ;

Step 3:
Agent performs an action: move.
Partial state:
Empty(3, 2) ;    Empty(3, 3) ;    Wall(3, 4) ;
Empty(4, 2) ;    Agent(4, 3, direction=(0, 1)) ; Wall(4, 4) ;
Marker(5, 2, quantity=1) ;        Empty(5, 3) ;    Wall(5, 4) ;

###45 steps are truncated.###

Step 49:
Agent performs a perception: frontIsClear. The result is False.
Partial state:
Wall(1, 0) ;     Empty(1, 1) ;    Empty(1, 2) ;
Wall(2, 0) ;     Agent(2, 1, direction=(0, -1)) ;          Empty(2,
    2) ;
Wall(3, 0) ;     Empty(3, 1) ;    Empty(3, 2) ;

The total step number is 105, the latter ones are truncated.

The total reward is 0.0

1. Depending on this information, please analyze the reason why
    the program failed to achieve 1.0 on this task variant and
    generate a new strategy to solve this task.
2. Generate 1 simple and short Python program according to the new
    strategy to tackle the task, avoid using comment.
```

```
3. Convert the Python program to the Karel dsl program.
```

## K.7 Agent and Program execution trace.

```
I'll provide you with the code you developed previously, with the
    goal of refining it. To guide your revision, you'll receive
    the specific task name and a description. Since there are 32
    different versions of the task that share the same objective
    but differ by random seeds, I will identify the specific
    variant where the performance of the program is most lacking.
    Additionally, you'll get the initial state of the task, the
    code, and a detailed trajectory demonstrating how the code
    operates within this particular scenario. This trajectory will
     detail each action step-by-step, indicate which section of
    your code is active, and show a localized snapshot of the
    environment (a 3x3 area centered on the agent) during
    execution. Rewards received by the agent will also be shown
    during these steps.

Task name: DOORKEY
Task map: The map is a 8x8 grid surrounded by walls that is
    vertically split into two chambers. The left chamber is 6x3
    grid and the right chamber is 6x2 grid. There is a marker
    placed randomly on the left chamber as a key, and another
    marker placed randomly on the right chamber as a goal.
Task agent position: The agent starts on a random cell on the left
     chamber facing east.
Task goal: The goal of the agent is to pick up a marker on the
    left chamber, which opens a door connecting both chambers.
    Allow the agent to reach and put a marker on the goal marker.
Task return: Picking up the first marker yields a 0.5 reward, and
    putting a marker on the goal marker yields an additional 0.5.

Initial state:
Wall(0, 0) ;     Wall(0, 1) ;     Wall(0, 2) ;     Wall(0, 3) ;
    Wall(0, 4) ;     Wall(0, 5) ;     Wall(0, 6) ;     Wall(0, 7) ;
Wall(1, 0) ;     Empty(1, 1) ;    Empty(1, 2) ;    Empty(1, 3) ;
    Wall(1, 4) ;     Empty(1, 5) ;    Marker(1, 6, quantity=1) ;
         Wall(1, 7) ;
Wall(2, 0) ;     Empty(2, 1) ;    Empty(2, 2) ;    Empty(2, 3) ;
    Wall(2, 4) ;     Empty(2, 5) ;    Empty(2, 6) ;    Wall(2, 7) ;
Wall(3, 0) ;     Empty(3, 1) ;    Empty(3, 2) ;    Empty(3, 3) ;
    Wall(3, 4) ;     Empty(3, 5) ;    Empty(3, 6) ;    Wall(3, 7) ;
Wall(4, 0) ;     Empty(4, 1) ;    Agent(4, 2, direction=(0, 1)) ;
    Empty(4, 3) ;    Wall(4, 4) ;     Empty(4, 5) ;    Empty(4, 6) ;
       Wall(4, 7) ;
Wall(5, 0) ;     Empty(5, 1) ;    Marker(5, 2, quantity=1) ;
    Empty(5, 3) ;    Wall(5, 4) ;     Empty(5, 5) ;    Empty(5, 6) ;
       Wall(5, 7) ;
Wall(6, 0) ;     Empty(6, 1) ;    Empty(6, 2) ;    Empty(6, 3) ;
    Wall(6, 4) ;     Empty(6, 5) ;    Empty(6, 6) ;    Wall(6, 7) ;
Wall(7, 0) ;     Wall(7, 1) ;     Wall(7, 2) ;     Wall(7, 3) ;
    Wall(7, 4) ;     Wall(7, 5) ;     Wall(7, 6) ;     Wall(7, 7) ;

Program:
def run():
    while noMarkersPresent():
        if frontIsClear():
```

```
            move()
        else:
            turnLeft()
            if frontIsClear():
                move()
                turnRight()
    pickMarker()
    while noMarkersPresent():
        if frontIsClear():
            move()
        else:
            turnLeft()
    putMarker()

The average reward on 32 task variants is:
0.5

Trajectory:

Step 1:
Program:
def run():
    while noMarkersPresent():  # Currently executing this line
        if frontIsClear():
            move()
        else:
            turnLeft()
            if frontIsClear():
                move()
                turnRight()
    pickMarker()
    while noMarkersPresent():
        if frontIsClear():
            move()
        else:
            turnLeft()
    putMarker()

Agent performs a perception: noMarkersPresent. The result is True.
Partial state:
Empty(3, 1) ;    Empty(3, 2) ;    Empty(3, 3) ;
Empty(4, 1) ;    Agent(4, 2, direction=(0, 1)) ; Empty(4, 3) ;
Empty(5, 1) ;    Marker(5, 2, quantity=1) ;       Empty(5, 3) ;

Step 2:
Program:
def run():
    while noMarkersPresent():
        if frontIsClear():  # Currently executing this line
            move()
        else:
            turnLeft()
            if frontIsClear():
                move()
                turnRight()
    pickMarker()
    while noMarkersPresent():
```

```
        if frontIsClear():
            move()
        else:
            turnLeft()
    putMarker()

Agent performs a perception: frontIsClear. The result is True.
Partial state:
Empty(3, 1) ;    Empty(3, 2) ;    Empty(3, 3) ;
Empty(4, 1) ;    Agent(4, 2, direction=(0, 1)) ; Empty(4, 3) ;
Empty(5, 1) ;    Marker(5, 2, quantity=1) ;        Empty(5, 3) ;

Step 3:
Program:
def run():
    while noMarkersPresent():
        if frontIsClear():
            move()  # Currently executing this line
        else:
            turnLeft()
            if frontIsClear():
                move()
                turnRight()
    pickMarker()
    while noMarkersPresent():
        if frontIsClear():
            move()
        else:
            turnLeft()
    putMarker()

Agent performs an action: move.
Partial state:
Empty(3, 2) ;    Empty(3, 3) ;    Wall(3, 4) ;
Empty(4, 2) ;    Agent(4, 3, direction=(0, 1)) ; Wall(4, 4) ;
Marker(5, 2, quantity=1) ;        Empty(5, 3) ;    Wall(5, 4) ;

###45 steps are truncated.###

Step 49:
Program:
def run():
    while noMarkersPresent():
        if frontIsClear():  # Currently executing this line
            move()
        else:
            turnLeft()
            if frontIsClear():
                move()
                turnRight()
    pickMarker()
    while noMarkersPresent():
        if frontIsClear():
            move()
        else:
            turnLeft()
    putMarker()
```

```
Agent performs a perception: frontIsClear. The result is False.
Partial state:
Wall(1, 0) ;    Empty(1, 1) ;    Empty(1, 2) ;
Wall(2, 0) ;    Agent(2, 1, direction=(0, -1)) ;          Empty(2,
    2) ;
Wall(3, 0) ;    Empty(3, 1) ;    Empty(3, 2) ;

The total step number is 105, the latter ones are truncated.

The total reward is 0.0

1. Depending on this information, please analyze the reason why
    the program failed to achieve 1.0 on this task variant and
    generate a new strategy to solve this task.
2. Generate 1 simple and short Python program according to the new
    strategy to tackle the task, avoid using comment.
3. Convert the Python program to the Karel dsl program.
```

## L    MINIGRID ENVIRONMENT

Minigrid (Chevalier-Boisvert et al., 2023) is a grid-world environment designed for conducting research on reinforcement learning (Guan et al., 2022; Jelley et al., 2023; Furelos-Blanco et al., 2023). Unlike the Karel environment (Pattis, 1981), Minigrid has a richer set of objects, and each object comes with a color. Besides Karel's actions, the Minigrid agent also can "toggle" objects, like unlocking a door or opening a box. The environment has built-in tasks that range from simple navigation to complex multi-step reasoning.

To evaluate our proposed LLM-GS framework on the Minigrid environment, we designed and implemented a domain-specific language (DSL), which contains more actions and object types than the Karel one. We will introduce the DSL in Appendix L.1, the tasks in Appendix L.2, the result in Appendix L.3, and the prompts in Appendix L.4.

### L.1    MINIGRID DSL

Here, we present the DSL we designed for the Minigrid environment. The actions are similar to the ones from Karel. The perceptions, on the other hand, are distinct from Karel. Specifically, the perception functions in our Minigrid DSL are parameterized, i.e., require an input parameter such as an object, since there are multiple object types in the domain, unlike the case in Karel. Thus, we implemented two novel perceptions: front_object_type( type ) and front_object_color( color ), which are completely different from the ones in Karel DSL. They check if the cell right in front of the agent has a certain type or a certain color object. We list the Python to DSL converting rules in Table 9 and the probabilities of the production rules in Table 10. The setting for the program mutation is the same as the one in Karel DSL.

Table 9: Python to DSL converting rules in Minigrid.

| Python | DSL |
| --- | --- |
| def run(): s | DEF run m( s m) |
| while b: s | WHILE c( b c) w( s w) |
| if b: s | IF c( b c) i( s i) |
| if b: s else: s | IFELSE c( b c) i( s i) ELSE e( s e) |
| for i in range(n): s | REPEAT R=n r( s r) |
| not h | not c( h c) |
| front_is_clear() | front_is_clear |
| front_object_type(type) | front_object_type h( type h) |
| front_object_color(color) | front_object_color h( color h) |
| is_carrying_object() | is_carrying_object |
| forward() | forward |
| left() | left |
| right() | right |
| pickup() | pickup |
| drop() | drop |
| toggle() | toggle |

### L.2    MINIGRID TASKS

We evaluate three tasks from the Minigrid environment: Lava Gap (Appendix L.2.1), Put Near (Appendix L.2.2), and Red Blue Door (Appendix L.2.3). We include an example illustration of the three tasks in Figure 21 and introduce the details of each task in the following sections.

Table 10: Probabilities of the production rules in Minigrid.

| Category | Rule | Probability |
|---|---|---|
| **Program P** | Statement | 1.0 |
| **Statement S** | While | 0.15 |
| | Repeat | 0.03 |
| | Concatenate | 0.5 |
| | If | 0.08 |
| | Ifelse | 0.04 |
| | Action | 0.2 |
| **Condition c** | Boolean | 0.5 |
| | not | 0.5 |
| **Action a** | forward | 0.5 |
| | left | 0.15 |
| | right | 0.15 |
| | pickup | 0.08 |
| | drop | 0.08 |
| | toggle | 0.04 |
| **Boolean b** | front_is_clear | 0.125 |
| | front_object_type | 0.5 |
| | front_object_color | 0.25 |
| | is_carrying_object | 0.125 |
| **Type t** | lava | 0.25 |
| | door | 0.25 |
| | ball | 0.25 |
| | box | 0.25 |
| **Color o** | blue | 0.5 |
| | red | 0.5 |
| **Number n** | i ($0 \leq i \leq 19$) | 0.05 |

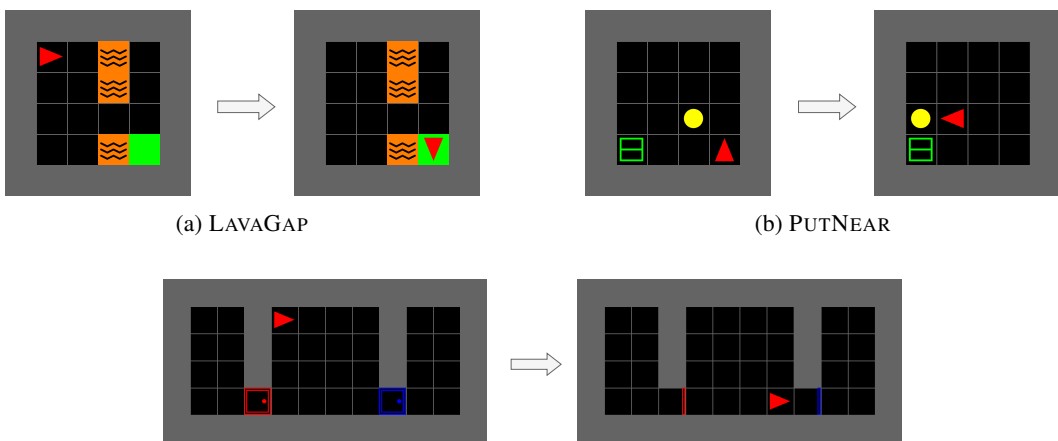

(a) LAVAGAP

(b) PUTNEAR

(c) REDBLUEDOOR

Figure 21: Illustrations of the initial and final state of the task we chose for the Minigrid (Chevalier-Boisvert et al., 2023) evaluation. The position of obstacles, objects, and the agent's position are randomly set according to the configurations of each task.

### L.2.1 LAVAGAP

In the task LavaGap, the agent must pass through a gap in a vertical strip of lava and reach the goal. Stepping on the lava will terminate the episode. We set the reward function of LavaGap to give a reward of -1.0 when stepping on lava and a reward of 1.0 when reaching the goal.

| Purpose | Prompt Text |
|---|---|
| Task Name | LAVAGAP |
| Map Description | The map is a 6x6 grid surrounded by walls. There is a vertical strip of lava with a randomly selected gap in the middle of the map and a goal square at the bottom right corner. |
| Initial Position | The agent starts at the top left corner of the map facing east. |
| Task Goal | The agent has to reach the goal square at the bottom right corner of the map without step on lava. Touching the lava terminates the episode. |
| Task Reward | If the agent reaches the goal, it will receive a reward of 1. If the agent steps on lava, it will receive a reward of -1. |

### L.2.2 PUTNEAR

In the task PutNear, the agent needs to pick up the *move* object and drop it next to the *target* object. We modified the task to fix the *move* object as a ball and the *target* object as a box. Picking up the *move* object will get a reward of 0.5 and dropping it next to the *target* object will receive the remaining 0.5. Picking up the wrong object will get a reward of -1.0.

| Purpose | Prompt Text |
|---|---|
| Task Name | PUTNEAR |
| Map Description | The map is a 6x6 grid surrounded by walls. There are two objects, one is a ball and the other is a box. The two objects are randomly placed on the map. |
| Initial Position | The agent starts at the top left corner of the map facing east. |
| Task Goal | The agent has to first locate and pick up the ball. Next, it needs to find the box and drop the ball either to the right or left of it. Picking up the wrong object will terminate the episode. |
| Task Reward | If the agent picks up the ball, it will receive a reward of 0.5. If the agent drops the ball to the right or the left of the box, it will receive a reward of 1. If the agent picks up the wrong object, it will receive a reward of -1. |

### L.2.3 REDBLUEDOOR

In the task RedBlueDoor, the agent must first open the red door and then the blue one. Opening the red door first will get a reward of 0.5, followed by an additional 0.5 for opening the blue door afterward. Opening the blue door before the red one will get a reward of -1.0.

| Purpose | Prompt Text |
|---|---|
| Task Name | REDBLUEDOOR |
| Map Description | The map is a 6x12 grid surrounded by walls that are vertically split into three chambers. The left chamber is a 4x2 grid, the middle chamber is a 4x4 grid and the right chamber is a 4x2 grid. On the wall between the left and middle chamber, there is a randomly placed red door. On the wall between the middle and right chamber, there is a randomly placed blue door. |
| Initial Position | The agent starts at a random position with a random direction in the middle chamber. |
| Task Goal | The agent has to first find and open the red door, then find and open the blue door. Opening the blue door first will terminate the episode. |
| Task Reward | If the agent opens the red door first, it will receive a reward of 0.5. If the agent opens the blue door after opening the red door, it will receive a reward of 0.5. If the agent opens the blue door first, it will receive a reward of -1. |

### L.3 MINIGRID DOMAIN

#### L.3.1 EVALUATION SETUP

We evaluate our proposed LLM-GS framework using the Minigrid tasks LAVAGAP, PUTNEAR, and REDBLUEDOOR. More task details can be found in Appendix L.2. LLM-GS is compared with the best-performing baseline HC. The evaluation metric is the same as the one mentioned in Section 5.1 except for $C = 16$ and $N = 10^4$. For the setup, we use GPT-4o (gpt-4o-2024-08-06 with temperature=1.0, top_p=1.0) as our LLM module to generate the initial search population. The scheduler of our proposed Scheduled HC starts from $K_{start} = 32$ to $K_{end} = 2048$. We evaluate our LLM-GS and HC with 8 random seeds.

#### L.3.2 EXPERIMENTAL RESULTS

Figure 22 presents the experimental result, indicating that our proposed LLM-GS is more sample-efficient compared to HC on all the tasks.

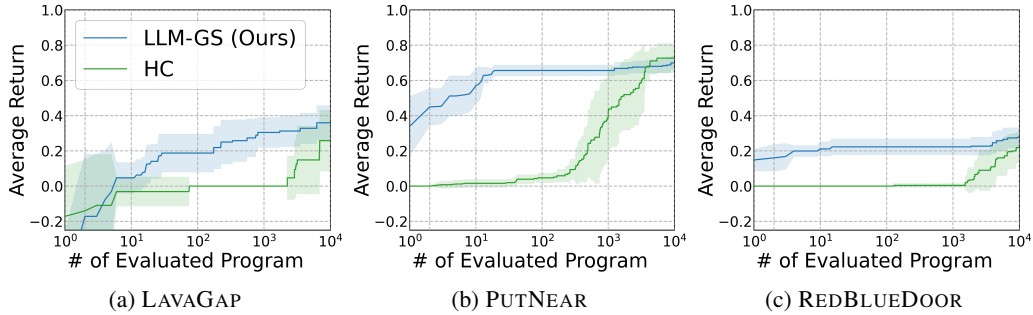

(a) LAVAGAP      (b) PUTNEAR      (c) REDBLUEDOOR

Figure 22: **Efficiency in Minigrid tasks.** We compare our proposed LLM-GS framework against best-performing baseline HC in three Minigrid tasks LAVAGAP, PUTNEAR, and REDBLUEDOOR. The results show that our LLM-GS is more sample-efficient than HC.

### L.4 MINIGRID PROMPTS

#### L.4.1 THE SYSTEM PROMPT

```
In the Minigrid environment, you navigate a grid-based world
    referred to as a "map." Within this map, an entity called the
    "agent" can move, change direction, and interact with objects.
     The map also includes obstacles like "walls," which block the
     agent's movement; if the agent encounters a wall, it remains
    stationary. Another obstacle is "lava," which ends the episode
     if the agent steps on it. Additionally, there are "doors"
    that block the agent until opened using the "toggle" action.

At the start of each episode, various objects are scattered across
     the map, and the agent can interact with them as needed.
    These objects fall into four categories: "lava," "door," "ball
    ," and "box." Except for lava, objects can appear in two
    colors: "red" and "blue."

Lava: Ends the episode when stepped on.
Door: Blocks the agent's path until opened using the "toggle"
    action.
Ball: Can be picked up and dropped.
Box: Can be picked up and dropped.

Your objective is to generate the appropriate Python program based
     on a given task name and description. This Python program
```

```
    will encompass actions enabling the agent to engage with the
    environment, alongside perceptions facilitating the agent's
    recognition of the environment's dynamics.

Here are the available actions for the agent:
forward(): Asks the agent to move forward one cell. The agent will
    stay still if a wall or an unopened door is blocking its way.
    The episode will be terminated if the agent steps on lava.
left(): Asks the agent to rotate 90 degrees counterclockwise.
right(): Asks the agent to rotate 90 degrees clockwise.
pickup(): Asks the agent to pick up an object right in front of
    the current cell. The agent can only carry one object at a
    time.
drop(): Asks the agent to drop the object it is carrying right in
    front of the current cell. The agent can only drop the object
    if it is carrying one and there is no object in the cell.
toggle(): Asks the agent to change the state of a door directly in
    front of its current position. The agent can use this action
    to open or close the door immediately ahead.

Here are the available perceptions of the agent:
front_is_clear(): Returns True if there is no wall and an unopened
    door right in front of the agent. It can only check wall and
    unopened door, so if there is lava right in front of the agent
    , it will return True.
front_object_type(type: Literal["lava", "door", "ball", "box"]):
    Returns True if there is an object of the specified type right
    in front of the agent. The type can be and only can be "lava
    ", "door", "ball", or "box".
front_object_color(color: Literal["red", "blue"]): Returns True if
    there is an object of the specified color right in front of
    the agent. The color can be and only can be "red" or "blue".
is_carrying_object(): Returns True if the agent is carrying an
    object.

There are some limitations to the Python program:
- do not define other functions besides run()
- do not call other functions
- do not define variables
- do not use True, False, break, continue, return, ==, !=, elif,
    or, and

Python to Minigrid dsl conversion
1. "def run(): s" to "DEF run m( s m)"
2. "while b: s" to "WHILE c( b c) w( s w)"
3. "if b: s" to "IF c( b c) i( s i)"
4. "if b: s else: s" to "IFELSE c( b c) i( s i) ELSE e( s e)"
5. "for i in range(n): s" to "REPEAT R=n r( s r)"
6. "not h" to "not c( h c)"
7. "front_is_clear()" to "front_is_clear"
8. "front_object_type(type)" to "front_object_type h( type h)"
9. "front_object_color(color)" to "front_object_color h( color h)"
10. "is_carrying_object()" to "is_carrying_object"
11. "forward()" to "forward"
12. "left()" to "left"
13. "right()" to "right"
14. "pickup()" to "pickup"
15. "drop()" to "drop"
16. "toggle()" to "toggle"
```

### L.4.2 THE USER PROMPT OF THE TASK LAVAGAP

```
I'll provide you with the task name and description.

Task name: LAVAGAP
Task map: The map is a 6x6 grid surrounded by walls. There is a
    vertical strip of lava with a randomly selected gap in the
    middle of the map and a goal square at the bottom right corner
    .
Task agent position: The agent starts at the top left corner of
    the map facing east.
Task goal: The agent has to reach the goal square at the bottom
    right corner of the map without step on lava. Touching the
    lava terminates the episode.
Task return: If the agent reaches the goal, it will receive a
    reward of 1. If the agent steps on lava, it will receive a
    reward of -1.

1. Generate 1 simple and short Python program to tackle the task,
    avoid using comments.
2. Convert the Python program to the Minigrid dsl program.
```

### L.4.3 THE SAMPLE RESPONSE OF LLM IN THE TASK LAVAGAP

```python
def run():
    while not front_object_type("lava"):
        forward()
    left()
    forward()
    right()
    while front_is_clear():
        forward()
    right()
    forward()
    left()
    while front_is_clear():
        forward()
```

```
Minigrid DSL program:
```
```
DEF run m(
    WHILE c( not c( front_object_type h( "lava" h) c) c) w(
        forward
    w)
    left
    forward
    right
    WHILE c( front_is_clear c) w(
        forward
    w)
    right
    forward
    left
    WHILE c( front_is_clear c) w(
        forward
    w)
m)
```

```
```
```

## M  LIMITATION

**Adapting to more complex environments beyond Karel.** We believe developing more realistic and complex programmatic RL environments is a promising future direction. However, to make a new environment suitable for programmatic reinforcement learning (PRL), we must carefully craft a domain-specific language (DSL) that defines action and perception primitives tailored to the environment's constraints. For example, in the `SpaceInvaders Atari` environment, actions might be defined as `Left, Right`, and `Fire`, while perceptions could include `AlienInFront, BulletInFront`, and `ObstacleInFront`. Only once such a DSL is in place can our framework be effectively implemented.

**Availability of Capable LLMs.** Our work assumes the availability of capable LLMs with common sense and programming skills. Training large language models (LLMs) from scratch is resource-intensive, posing significant financial and computational costs. The process requires vast amounts of high-quality data and cutting-edge hardware like GPUs or TPUs.

**Necessity of domain experts.** We need a domain expert who understands both the basic grammar of the domain-specific language (DSL) and the domain of interest, including its low-level action and perception primitives and simple control flows. This expert will provide a system prompt that explains the environmental and domain concepts to the LLM when introducing it to a new environment. That said, adopting our framework to a new domain requires such an expert to provide domain prompts. Note that the task prompt, describing the goal of tasks, is easy to write and accessible to general users, as it simply converts task documentation into a natural language description.

