# OpenReview forum: "Synthesizing Programmatic Reinforcement Learning Policies with Large Language Model Guided Search"
_ICLR.cc/2025/Conference — ICLR 2025 Poster_

### Official Review · Reviewer_CapU · 2024-11-02

**Soundness:** 2
**Presentation:** 3
**Contribution:** 3
**Rating:** 6
**Confidence:** 4

**Summary:**

Programmatic policies for reinforcement learning have gained popularity in recent years, with state-of-the-art methods often employing search techniques that utilize various heuristics and initialization strategies. Given that these search methods are typically localized and proximity-based, the choice of initialization point becomes crucial. The authors present a novel LLM-based search algorithm, LLM-GS, which uses large language models (LLMs) for initializing the search, capitalizing on their coding capabilities. They acknowledge that LLMs often struggle to generate effective initializations within the domain-specific language (DSL) of the problem and introduce a Pythonic-DSL approach to address this challenge. Additionally, they propose a new search algorithm called scheduled hill climbing, which optimizes the search budget by starting with a smaller allocation and gradually increasing it as the search progresses. Through experiments in the Karel the Robot domain, they demonstrate the superiority of their algorithm compared to several search baselines and conduct thorough ablation studies to assess the impact of each contribution on performance.

**Strengths:**

* The authors have identified a significant limitation in existing search algorithms for programmatic reinforcement learning: their localized nature and the crucial role of initialization. They effectively combine this insight with the proven capabilities of LLMs in coding tasks to develop a straightforward yet powerful algorithm, LLM-GS. Furthermore, they propose a search strategy that aligns with their initialization approach, recognizing that they are likely starting from a strong position and can save their search budget for later stages.
* They demonstrate the superior performance of LLM-GS compared to several search baselines, instilling confidence in its effectiveness, particularly in the Karel domain.
* The ablation studies are well-structured and thorough, effectively investigating each component to clarify the sources of performance improvements.
* The authors address potential concerns regarding LLM memorization and data leakage, which adds to the credibility of their results.
* I found the LLM Revision experiment in Section 5.5 particularly fascinating, as it explores whether LLMs can be leveraged more effectively in the search process to refine the initially proposed program. This experiment highlights that existing simple techniques may not perform as well as anticipated.

**Weaknesses:**

I have two main concerns with the paper that prevent me from giving it a higher score:
* The authors have conducted experiments solely in the Karel the Robot domain. Testing the LLM-GS algorithm in another domain, such as MicroRTS, would enhance confidence in its effectiveness across different problem domains and DSLs. **Addressing this issue could lead me to reconsider my score.**
* The ablation study comparing search algorithms is quite limited, especially given the close results shown in Figure 6. It appears that the LLM provides such a strong starting point that the specific search algorithm has minimal impact. All search algorithms perform reasonably well on CleanHouse, and the results are very similar among the top 2-3 algorithms on DoorKey. Including results from additional domains would help establish the significance of the search algorithm. Furthermore, the scheduled hill climbing algorithm involves several design choices, such as the sinusoidal schedule and logarithmic interpolation; conducting separate ablation studies on the importance of each of these choices could be beneficial.

**Questions:**

* Do the equations for the scheduled hill climbing algorithm derive from prior work, or is there additional intuition apart from the decision to increase the search budget as the algorithm progresses? There are several key choices here—such as the logarithmic interpolation and sinusoidal schedule—that don’t seem entirely intuitive and are introduced without sufficient justification. Could you provide more details on these decisions?
* Are the program embeddings used for latent space search generated by the LLM?

---

> ### Author Response · Authors · 2024-11-27
> **Response to Reviewer CapU (Part 1/2)**
>
> We sincerely thank the reviewer for the thorough and constructive comments. Please find the response to your questions below.
>
> > The authors have conducted experiments solely in the Karel the Robot domain. Testing the LLM-GS algorithm in another domain, such as MicroRTS, would enhance confidence in its effectiveness across different problem domains and DSLs. Addressing this issue could lead me to reconsider my score.
>
> Due to the limited time we have during the rebuttal period, we could not conduct experiments in the MicroRTS domain since running each method against CoacAI on map basesWorkers8x8A, one of the simplest tasks in MicroRTS, could take up to 16 hours per seed for search-based methods like HC so it’s more than 10 days in total and it takes even longer for RL based method like HPRL.
>
> Therefore, to include an additional domain, we adopt **the Minigrid domain** [1]. We designed the DSL for Minigrid to be similar to the MicroRTS DSL used by Moraes et al. [2]. Specifically, both the perception functions in our Minigrid DSL and the MicroRTS DSL are parameterized, i.e., require an input parameter such as an object, since there are multiple object types in the two domains, unlike the case in Karel.
>
> We present the results of three Minigrid tasks in Section 5.5 in the revised paper. As shown in Figure 9, **LLM-GS demonstrates better sample efficiency compared to HC, the best-performing baseline in the Karel domain**. For further details of the Minigrid experiments, please refer to Section 5.5 and Appendix L.
>
> > The ablation study comparing search algorithms is quite limited, especially given the close results shown in Figure 6. It appears that the LLM provides such a strong starting point that the specific search algorithm has minimal impact. All search algorithms perform reasonably well on CleanHouse, and the results are very similar among the top 2-3 algorithms on DoorKey. Including results from additional domains would help establish the significance of the search algorithm.
>
> In Figure 6, it is evident that HC-32 and HC-2048 excel only in specific tasks: HC-32 performs exceptionally well in CleanHouse, while HC-2048 stands out in DoorKey. In contrast, our proposed scheduled HC ranks among the best-performing methods in both DoorKey and CleanHouse, showcasing its robustness and efficiency. **That said, given a novel task of interest, we can confidently run our method LLM-GS instead of trying HC with different hyperparameters**.
>
> > Furthermore, the scheduled hill climbing algorithm involves several design choices, such as the sinusoidal schedule and logarithmic interpolation; conducting separate ablation studies on the importance of each of these choices could be beneficial.
>
> We thank the reviewer for the suggestion. There are three components in the proposed scheduled hill climbing (SHC): logarithmic interpolation, sinusoidal schedule, and logarithmic ratio. As suggested by the reviewer, **we have conducted additional experiments ablating each component**. We substitute them with their linear counterpart and testify all 8 variants in DoorKey and CleanHouse. The results shown in Appendix E in the revised paper indicate no significant differences between all the SHC variants, showcasing its robustness in design.
>
> > Do the equations for the scheduled hill climbing algorithm derive from prior work, or is there additional intuition apart from the decision to increase the search budget as the algorithm progresses? There are several key choices here—such as the logarithmic interpolation and sinusoidal schedule—that don’t seem entirely intuitive and are introduced without sufficient justification. Could you provide more details on these decisions?
>
> Our intuition to logarithmic interpolation and logarithmic ratio aims to allocate the most appropriate neighborhood size, k, to tasks of varying difficulties. The logarithm of the number of executed programs provides an indication of task difficulty, with the intuition that the optimal k should increase exponentially according to the structure of AST. For the sinusoidal schedule, we prioritize maintaining a stable neighborhood size k during both the early and final stages of the training process. We thank the reviewer for raising this question, and we have revised the paper to include these design intuitions in Appendix E.
>
> > Are the program embeddings used for latent space search generated by the LLM?
>
> No. We do not use LLMs to generate the program embeddings; instead, **the program embeddings are derived from Trivedi et al. [3]**. Their method trains a Variational Autoencoder (VAE) to produce a program embedding space from programs and their execution traces.

---

> ### Author Response · Authors · 2024-11-27
> **Response to Reviewer CapU (Part 2/2)**
>
> **References**
>
> [1] Maxime Chevalier-Boisvert, Bolun Dai, Mark Towers, Rodrigo Perez-Vicente, Lucas Willems, Salem Lahlou, Suman Pal, Pablo Samuel Castro, and Jordan Terry. Minigrid & miniworld: Modular & customizable reinforcement learning environments for goal-oriented tasks. In Neural Information Processing Systems, 2023.
>
> [2] Rubens O. Moraes, David S. Aleixo, Lucas N. Ferreira, and Levi H. S. Lelis. Choosing well
> your opponents: How to guide the synthesis of programmatic strategies. In Proceedings of the
> Thirty-Second International Joint Conference on Artificial Intelligence, International Joint Conference on Artificial Intelligence, 2023.
>
> [3] Dweep Trivedi, Jesse Zhang, Shao-Hua Sun, and Joseph J Lim. Learning to synthesize programs as interpretable and generalizable policies. In Neural Information Processing Systems, 2021.

---

> > ### Author Response · Authors · 2024-11-29
> > **Reminder: The reviewer-author discussion period ends in four days**
> >
> > We would like to express our sincere gratitude to the reviewer for the thorough and constructive feedback. We are confident that our responses adequately address the concerns and questions raised by the reviewer, including the following points:
> >
> > - **Additional results in a new domain (Minigrid with three tasks)**: Section 5.5 and Appendix L
> > - **An explanation of the result in Figure 6**
> > - **An additional ablation study and an explanation of the design intuitions of SHC**: Appendix E
> > - **A clarification of program embedding space**
> >
> > Please kindly let us know if the reviewer has any additional concerns or if further experimental results are required. We are fully committed to resolving any potential issues, should time permit. Again, we thank the reviewer for all the detailed review and the time the reviewer put into helping us to improve our submission.

---

> > > ### Comment · Reviewer_CapU · 2024-12-02
> > > **Thank you for the responses (score increased)**
> > >
> > > Thank you for your thorough responses to my questions regarding the scheduled HC algorithm and the ablation studies, as well as for running experiments in an additional domain. I believe incorporating MicroRTS before the camera-ready version would significantly strengthen your paper, but I am already convinced by the results and methodology. As such, I’m happy to increase my score now.

---

> > > > ### Author Response · Authors · 2024-12-02
> > > >
> > > > We sincerely thank the reviewer for acknowledging our rebuttal and for providing insightful feedback, which has significantly strengthened our submission.

---

### Official Review · Reviewer_C8r6 · 2024-11-03

**Soundness:** 3
**Presentation:** 3
**Contribution:** 3
**Rating:** 8
**Confidence:** 5

**Summary:**

In this paper, the authors propose a framework for PRL (Programmatic Reinforcement Learning) based on a new search algorithm in program space (SHC), whose initial population of programs  is generated from a capable LLM rather than obtained via random generation.
The authors' main contributions are a prompting strategy which correctly outlines each PRL task to the LLM, coupled with the idea of having the LLM generate both Python code and the DSL in which the solution to a given PRL task is supposed to be expressed; this allows the authors to get the LLM to generate programs with the correct DSL syntax, despite the DSL not having presumably been experienced by the LLM  during training.

**Strengths:**

The paper's idea is quite simple but effective, and one can see that it significantly improves over SOTA methods when it comes to sample efficiency/number of program evaluations during search, and for some tasks, also in terms of final average return. The paper is written clearly and features comprehensive ablation studies.

**Weaknesses:**

It is not quite clear which one of the two contributions (SHC and its initialisation with LLM-generated programs) is the one which contributes the most to the final result. From figure 4, it does not appear that the SHC algorithm is really that much more efficient than HC. Plus, the role of the Python -> DSL parser is not quite clear.

**Questions:**

I have two main points relating to the weaknesses mentioned above, which I would like to see addressed before possibly raising my score:
- The authors should run two more ablation studies of their LLM-GS method; one in in which they remove the LLM generated initialisation for SHC, and one in in which they run SHC which the initialisation strategy used by the SOTA HC. This would help address the weakness mentioned above.
- What is exactly the role of the parser? Does the LLM generate both Python and DSL code natively, or does it only generate Python code to be processed by the parser? Did the authors implemenent the parser themselves, of did they use an off-the-shelf parser?

More but less pressing questions:
- The advantage of SHC in terms of # of program evaluations is manifest. However, how large is its advantage in terms of simple walltime/wait time?
- HC (and therefore SHC as well) is basically a greedy search algorithm. Isn't there a strong possibility of it getting stuck in local maxima? To me this seems to be a bigger problem of HC compared to its sample inefficiency.
- Most of the discussion in section 3 (aside from the outline of the Karel domain) feels more like a Related Work section. I suggest that the authors move it to Related Work or to the Appendix.
- Which initialisation do the authors use for the ablation study in figure 6? To they use LLM-generated programs or randomly generated ones?
 - How is the experiment in section 5.5 conducted? Is the LLM asked to revise programs, and these are then used as initialisation for SHC? Or is SHC completely absent from the experiment?

---

> ### Author Response · Authors · 2024-11-27
> **Response to Reviewer C8r6**
>
> We sincerely thank the reviewer for the thorough and constructive comments. Please find the response to your questions below.
>
> > The authors should run two more ablation studies of their LLM-GS method; one in in which they remove the LLM generated initialisation for SHC, and one in in which they run SHC which the initialisation strategy used by the SOTA HC. This would help address the weakness mentioned above.
>
> We completely agree with the reviewer that comparing different initializations to the SHC search is essential. Hence, we have shown this ablation study in **Figure 7 in the original paper**, where we compared two initialization strategies: (1) **LLM initialization** which uses LLM-generated programs as proposed in this work, and (2) **random initialization** used by SOTA HC. The result shows that our proposed LLM initialization significantly outperforms random initialization in terms of sample efficiency.
>
> > What is exactly the role of the parser? Does the LLM generate both Python and DSL code natively, or does it only generate Python code to be processed by the parser? Did the authors implemenent the parser themselves, of did they use an off-the-shelf parser?
>
> To generate a program, we prompt the LLM to write the program in Python and then translate it into DSL, all in one response. We have included a sample output from the LLM in Appendix D.3.
>
> To translate an LLM-generated program to an abstract syntax tree (AST), we use an off-the-shelf Karel parser which can convert a string in DSL into an AST. We implemented a translator to convert a Python program into Karel DSL and fix some minor errors made by the LLM (rules listed in Table 6). We have updated Appendix B to make it clear.
>
> > The advantage of SHC in terms of # of program evaluations is manifest. However, how large is its advantage in terms of simple walltime/wait time?
>
> As suggested by the reviewer, we have additionally conducted **a wall time evaluation comparing different methods (HC, CEBS, CEM, and LLM-GS)** in DoorKey. The result is presented in Appendix I (Figure 18), showing that our proposed **LLM-GS framework is significantly more runtime efficient compared to all the baselines**, surpassing the baselines in less than 50 seconds.
>
> > HC (and therefore SHC as well) is basically a greedy search algorithm. Isn't there a strong possibility of it getting stuck in local maxima? To me this seems to be a bigger problem of HC compared to its sample inefficiency.
>
> As pointed out by the reviewer, greedy search algorithms can indeed get stuck in local maxima. Thus, these algorithms often have a restart strategy - to search programs from another initial point. In the cases of HC and SHC, after searching the neighborhood programs for k times without improving,  it switches to a new initial point and restarts the search process. In HC, the neighborhood size k is fixed, whereas in SHC, it can grow over time. **Hence the main contribution of our work is to utilize LLM-generated programs as restart initial points given LLM’s common sense and programming skills to potentially escape local maxima**.
>
> > Most of the discussion in section 3 (aside from the outline of the Karel domain) feels more like a Related Work section. I suggest that the authors move it to Related Work or to the Appendix.
>
> As suggested by the reviewer, we have moved the discussion of CEM and CEBS to Appendix F, while keeping the search space and HC in Section 3 since they are essential for understanding our proposed framework.
>
> > Which initialisation do the authors use for the ablation study in figure 6? To they use LLM-generated programs or randomly generated ones?
>
> The ablation study in Figure 6 uses LLM-generated programs. To make this clear, we have revised the caption of Figure 6 by adding “using LLM-initialized programs.”
>
> > How is the experiment in section 5.5 conducted? Is the LLM asked to revise programs, and these are then used as initialisation for SHC? Or is SHC completely absent from the experiment?
>
> We conduct this experiment to test whether LLMs can progressively revise and improve their generated programs, **so SHC and any other search methods are completely absent from this experiment**.
>
> We ask the LLM to repeatedly generate new programs given feedback from previously produced ones and observe that the performance saturates within a few rounds. This indicates the necessity of combining the advantages of LLM and the search method to achieve better performance. We have clarified this in Section 5.6 in the revised paper by adding “LLM itself without the help of search algorithms.”

---

> > ### Author Response · Authors · 2024-11-29
> > **Reminder: The reviewer-author discussion period ends in four days**
> >
> > We would like to express our sincere gratitude to the reviewer for the thorough and constructive feedback. We are confident that our responses adequately address the concerns and questions raised by the reviewer, including the following points:
> >
> > - **A clarification of Figure 7 caption**
> > - **An explanation of the parser**: Appendix D.3
> > - **An evaluation of wall time**: Appendix I (Figure 18)
> > - **A discussion of escaping local maxima with LLM-initialized restart programs**
> > - **A rearrangement of Section 3**: the discussion of CEM and CEBS has been moved to Appendix F
> > - **A clarification of Figure 6 caption**
> > - **A clarification of LL revision**: Section 5.6
> >
> > Please kindly let us know if the reviewer has any additional concerns or if further experimental results are required. We are fully committed to resolving any potential issues, should time permit. Again, we thank the reviewer for all the detailed review and the time the reviewer put into helping us to improve our submission.

---

> > > ### Author Response · Authors · 2024-12-03
> > > **Reminder: The reviewer-author discussion period ends in 10 hours**
> > >
> > > The deadline for reviewers to post a message to the authors is in 10 hours (Dec 2 23:59 AoR). We look forward to hearing from the reviewer.

---

> > > > ### Comment · Reviewer_C8r6 · 2024-12-03
> > > >
> > > > I thanks the authors for their diligent and detailed reply to my comments. Since all of my concerns have been addressed, I will raise my score to a straight accept.

---

> > > > > ### Author Response · Authors · 2024-12-03
> > > > >
> > > > > We are very grateful to the reviewer for acknowledging our rebuttal and for the effort the reviewer put into helping us improve our submission.

---

### Official Review · Reviewer_st6i · 2024-11-06

**Soundness:** 2
**Presentation:** 3
**Contribution:** 2
**Rating:** 6
**Confidence:** 4

**Summary:**

This paper studies learning programmatic-actions for RL, through heuristic search methods such as hill climbing. The paper focuses on the initialization problem. Rather than randomly initializing programs, the authors propose using LLM guided programs where the environment is described in natural language and a GPT-based model is prompted to bootstrap a set of programs. These programs are further improved using heuristic methods based on environment feedback. The authors also propose a scheduler for the hill climbing that allocates budget based on a sinusoidal scheduling. Finally, they find that generating DSLs directly is challenging due to domain gap; instead, they propose python-dsl where a python program is generated by the LLM and converted into DSL using rules. The resulting initialization and heuristic lead to improvement in sample complexity of the heuristics in a toy RL navigation benchmark.

**Strengths:**

Update: Authors addressed my concerns with many additional experiments. I increased my score accordingly.

1. The paper proposes LLM-based initialization for heuristic search methods. This is applicable to broader range of domains.

2. Experimental results indicate sample efficiency.

**Weaknesses:**

My main concerns are lack of realistic domains, novelty of the proposed idea, and significance of the experimental comparison.

1. The proposed idea is only tested on a toy RL benchmark which only add marginal practical value. Are there no other environments where programs can be actions and your idea can be applied?

2. In Figure-4, HC and your method compare very similarly. In fact, confidence intervals overlap in many of the tasks. It is not clear if the your method is significantly better. Can you conduct a statistical test to better compare these two methods?

3. Related to (2), HC learning curve is steeper than yours. Can you explain why?

4. While the complexity focus is mainly on number of programs, the cost of running a LLM is ignored. Can you add the cost of initialization to your plots to understand if spending more “flops” is in general better?

5. What is task variance? Is it random initialization in environment state or policy pool?

6. What happens if you initialize other heuristics with LLM-based programs? Do they compare favorably compared to your Scheduling-HC?

7. In line 425, it should be 500k.

**Questions:**

Please see above for specific questions.

---

> ### Author Response · Authors · 2024-11-27
> **Response to Reviewer st6i (Part 1/3)**
>
> We sincerely thank the reviewer for the thorough and constructive comments. Please find the response to your questions below.
>
> > The proposed idea is only tested on a toy RL benchmark which only add marginal practical value. Are there no other environments where programs can be actions and your idea can be applied?
>
> We follow existing programmatic RL works [1-3] and use the Karel domain as our testbed for evaluation. The Karel tasks characterize diverse aspects that are essential in solving various RL problems as discussed [1-2], including:
> - **Exploration**: to navigate the agent through maps to novel states that may be far from initial locations like Maze and CleanHouse.
> - **Complexity**: to perform specific actions at specific locations i.e., put or pick markers on marker grids like TopOff and Harvester.
> - **Multi-stage exploration**: to exhibit specific behaviors depending on the current stages like DoorKey.
> - **Additional Constraints**: to perform specific actions under restrictions e.g., traverse the environment without revisiting the same position like OneStroke, and place exactly one marker on all grids like Seeder.
>
> As suggested by the reviewer, **we have additionally explored adapting our framework to a new domain, the Minigrid domain [4], to showcase the adaptability of our proposed framework**, which simply requires defining sets of perceptions and actions for the domain of interest. We utilized the same control flows (FOR, WHILE…) as in the Karel DSL and chose actions as the Minigrid built-in ones. Unlike the Karel environment, Minigrid has multiple objects and colors, and it is necessary to identify them to solve the tasks. Therefore, we designed our perceptions to be parameterized, i.e., the perceptions may require input parameters, such as an object or a color.
>
> We present the results of three Minigrid tasks in Section 5.5. As shown in Figure 9, LLM-GS demonstrates better sample efficiency compared to HC, the best-performing baseline in the Karel domain. For further details of the Minigrid experiments, please refer to Section 5.5 and Appendix L.
>
> > In Figure-4, HC and your method compare very similarly. In fact, confidence intervals overlap in many of the tasks. It is not clear if the your method is significantly better. Can you conduct a statistical test to better compare these two methods?
>
> We thank the reviewer for the suggestion. To provide a statistical test and better compare the two methods, we use **the Student’s t-test [5]** to evaluate the statistical significance of our proposed LLM-GS compared to HC in terms of sample efficiency. We present this additional result in Appendix J (Figure 19) in the revised paper. **The result shows that LLM-GS outperforms HC in sample efficiency with statistical significance on 8 out of 10 Karel tasks, and the two methods perform comparably with the other two tasks.**

---

> ### Author Response · Authors · 2024-11-27
> **Response to Reviewer st6i (Part 2/3)**
>
> > Related to (2), HC learning curve is steeper than yours. Can you explain why?
>
> Our motivation is to improve the **sample efficiency** of programmatic RL, i.e., maximizing the expected return using as few program evaluations as possible, which aligns with the standard RL objective that aims to minimize the number of environment interactions. That said, we analyze the plots from the following aspects:
> - **Reaching a target average return with varying numbers of program evaluations**  (drawing a horizontal line in the plots): Table 2 below shows the number of program evaluations required for each method to reach an average return of 0.5. The result shows that LLM-GS requires fewer program evaluations to achieve an average return of 0.5 in all the tasks compared to HC.
> - **Average return given the same number of program executions** (drawing a vertical line in the plots): Table 1 below shows the average return when each method uses $k \in \{100, 1000\}$ program evaluations. LLM-GS outperforms HC in all the tasks using 100 programs. When using 1k programs, LLM-GS achieves a better average return in six Karel tasks compared to HC, and the two methods perform comparably in the rest of the four tasks. That said, LLM-GS is more sample-efficient than HC when the 1000 program elapses.
>
> | Table 1 | CleanHouse | DoorKey | FourCorners | Harvester | Maze | OneStroke | Seeder | Snake | StairClimber | TopOff |
> | --- | --- | --- | --- | --- | --- | --- | --- | --- | --- | --- |
> | LLM-GS | **189** | **6** | **5** | **3** | **1** | **2** | **3** | **73521** | **3** | **7** |
> | HC | 261 | 10454 | 1443 | 136 | 44 | 34 | 976 | 132989 | 323 | 1365 |
>
> | Table 2 | CleanHouse | DoorKey | FourCorners | Harvester | Maze | OneStroke | Seeder | Snake | StairClimber | TopOff |
> | --- | --- | --- | --- | --- | --- | --- | --- | --- | --- | --- |
> | LLM-GS (100 program evaluations) | **0.40** | **0.69** | **1.00** | **0.94** |**1.00** |**0.82** | **0.92** | **0.09** | **1.00** | **0.99** |
> | HC (100 program evaluations) | 0.25 | 0.17 | 0.09 | 0.41 | 0.74 | 0.73 | 0.15 | 0.05 | 0.25 | 0.04 |
> | LLM-GS (1000 program evaluations) | **0.96** | **0.86** | **1.00** |**0.97** | **1.00** | **0.90** | **0.97** | 0.13 | **1.00** | **1.00** |
> | HC (1000 program evaluations) | 0.90 | 0.40 | 0.35 | 0.88 | 1.00 | **0.90** | 0.51 | **0.14** | **1.00** | 0.43 |
>
> Finally, we would like to clarify that steepness is not often considered an important factor when analyzing sample-efficiency plots. We believe that **HC curves look steeper because their initial performance (random initializations) is much worse than LLM-GS with initialized programs**. As a result, there is more room for improvement for HC. Therefore, we believe **the steeper curves of HC should not be considered as an advantage**.
>
> > While the complexity focus is mainly on number of programs, the cost of running a LLM is ignored. Can you add the cost of initialization to your plots to understand if spending more “flops” is in general better?
>
> The standard reinforcement learning (RL) problem setting often considers interacting with environments to be expensive, dangerous (e.g., robotics), or even impossible (e.g., offline RL). Therefore, the development of RL algorithms focuses on minimizing the number of interactions required to get reasonable performance. In the case of programmatic RL, we consider the number of programs executed to measure the sample efficiency.
>
> As requested by the reviewer, we have conducted additional investigations on the cost of LLMs. Since we use a proprietary language model (GPT4) in our experiments, we cannot calculate the exact FLOPs used. Hence, **we estimate the cost of running the LLM in the following two aspects**:
> - **Time**: We provide a plot in Appendix I (Figure 18) illustrating the wall-clock time evaluation for the DoorKey task. The result shows that although LLM takes some time to generate the initial programs, it quickly surpasses other baselines in a few seconds, highlighting the efficiency of our proposed method in terms of real-time elapsed.
> - **Money**: We use 48 API calls to initialize the search population of 32 programs, which costs approximately USD 0.50 when using GPT-4 (gpt-4-turbo-2024-04-09).

---

> ### Author Response · Authors · 2024-11-27
> **Response to Reviewer st6i (Part 3/3)**
>
> > What is task variance? Is it random initialization in environment state or policy pool?
>
> **Each task's variance arises from the randomness in the environment's initial states**, including the agent's position and direction and the placement of markers and walls. We thank the reviewer for raising this question, and we have updated Section 5.1 (the Metric paragraph) to make this clear.
>
> > What happens if you initialize other heuristics with LLM-based programs? Do they compare favorably compared to your Scheduling-HC?
>
> We completely agree with the reviewer that comparing different search methods with the same set of LLM-initialized is essential. Therefore, **Section 5.3 (Figure 6) in the original paper exactly presents this comparison, where we initialize the CEM, CEBS, HC, and Scheduled HC heuristics with LLM-generated programs** in two tasks: DoorKey and CleanHouse. The result shows that our proposed scheduled HC is among the best-performing heuristics in both tasks. We have revised the paper to clarify this in the caption of Figure 6 by adding “using LLM-initialized programs.”
>
> > In line 425, it should be 500k.
>
> We thank the reviewer for pointing this out, and we have fixed this in our revised paper.
>
> **References**
>
> [1] Dweep Trivedi, Jesse Zhang, Shao-Hua Sun, and Joseph J Lim. Learning to synthesize programs as interpretable and generalizable policies. In Neural Information Processing Systems, 2021.
>
> [2] Guan-Ting Liu, En-Pei Hu, Pu-Jen Cheng, Hung-Yi Lee, and Shao-Hua Sun. Hierarchical programmatic reinforcement learning via learning to compose programs. In International Conference on Machine Learning, 2023.
>
> [3] Tales Henrique Carvalho, Kenneth Tjhia, and Levi Lelis. Reclaiming the source of programmatic policies: Programmatic versus latent spaces. In International Conference on Learning Representations, 2024.
>
> [4] Maxime Chevalier-Boisvert, Bolun Dai, Mark Towers, Rodrigo Perez-Vicente, Lucas Willems, Salem Lahlou, Suman Pal, Pablo Samuel Castro, and Jordan Terry. Minigrid & miniworld: Modular & customizable reinforcement learning environments for goal-oriented tasks. In Neural Information Processing Systems, 2023
>
> [5] Student. The probable error of a mean. Biometrika, 6(1):1–25, 03 1908. ISSN 0006-3444

---

> > ### Author Response · Authors · 2024-11-29
> > **Reminder: The reviewer-author discussion period ends in four days**
> >
> > We would like to express our sincere gratitude to the reviewer for the thorough and constructive feedback. We are confident that our responses adequately address the concerns and questions raised by the reviewer, including the following points:
> >
> > - **Additional results in a new domain (Minigrid with three tasks)**: Section 5.5 and Appendix L
> > - **A statistical test of our main result**: Appendix J (Figure 19)
> > - **A discussion of HC's learning curves' steepness**
> > - **An estimation of the cost of running an LLM**
> > - **A clarification of task variance**: Section 5.1 (the Metric paragraph)
> > - **A clarification of Figure 6 caption**
> >
> > Please kindly let us know if the reviewer has any additional concerns or if further experimental results are required. We are fully committed to resolving any potential issues, should time permit. Again, we thank the reviewer for all the detailed review and the time the reviewer put into helping us to improve our submission.

---

> > > ### Author Response · Authors · 2024-12-03
> > > **Reminder: The reviewer-author discussion period ends in 10 hours**
> > >
> > > The deadline for reviewers to post a message to the authors is in 10 hours (Dec 2 23:59 AoE). We look forward to hearing from the reviewer.

---

> > > > ### Author Response · Authors · 2024-12-03
> > > > **Still looking forward to the reviewer's feedback**
> > > >
> > > > Since the deadline for reviewers to post a message has just passed, the reviewer won't be able to post an official comment anymore. **We would greatly appreciate it if the reviewer could edit the original review to let us know if our rebuttal and the revised paper sufficiently address the questions and concerns raised by the reviewer**. This would be extremely helpful for us in improving our submission.

---

### Meta-Review · Area_Chair_F9VF · 2024-12-21

**Metareview:**

The paper presents a mechanism to use LLMs to find policies in a Domain Specific Language (DSL) for the Programmatic Reinforcement Learning paradigm. The technique builds on the Hill Climbing (HC) algorithm that effectively searches for programmatic policies directly in the program space. The proposed technique provides a mechanism to initialize the HC algorithm with promising candidates instead of with random programs. The experiments show the efficacy of the proposed technique in a toy RL navigation benchmark.

**Additional Comments On Reviewer Discussion:**

The authors provided detailed responses to the concerns raised by the reviewers. There was a consensus among the reviewers that the paper should be accepted.

---

### Decision · Program_Chairs · 2025-01-22

Accept (Poster)